# Impacts of combined microphysical and land-surface uncertainties on convective clouds and precipitation in different weather regimes

Christian Barthlott[1], Amirmahdi Zarboo[1], Takumi Matsunobu[2], and Christian Keil[2]

[1]Institute of Meteorology and Climate Research (IMK-TRO), Department Troposphere Research, Karlsruhe Institute of Technology (KIT), Karlsruhe, Germany
[2]Meteorologisches Institut, Ludwig-Maximilians-Universität, Munich, Germany

**Correspondence:** Christian Barthlott (christian.barthlott@kit.edu)

**Abstract.** To reduce the underdispersion of precipitation in convective-scale ensemble prediction systems, we investigate the relevance of microphysical and land-surface uncertainties for convective-scale predictability. We use three different initial soil moisture fields and study the response of convective precipitation to varying cloud condensation nuclei (CCN) concentrations and different shape parameters of the cloud droplet size distribution (CDSD) by applying a novel combined-perturbation strategy. Using the new icosahedral nonhydrostatic ICON model, we construct a 60-member ensemble for cases with summertime convection under weak and strong synoptic-scale forcing over central Europe. We find a systematic positive soil moisture–precipitation feedback for all cases, regardless of the type of synoptic forcing and a stronger response of precipitation to different CCN concentrations and shape parameters for weak forcing than for strong forcing. While the days with weak forcing show a systematic decrease in precipitation with increasing aerosol loading, days with strong forcing also show nonsystematic responses for some values of the shape parameters. The large magnitude of precipitation deviations compared to a reference simulation ranging between $-23\%$ to $+18\%$ demonstrates that the uncertainties investigated here and, in particular, their collective effect are highly relevant for quantitative precipitation forecasting of summertime convection in central Europe. A rain water budget analysis is used to identify the dominating source and sink terms and their response to the uncertainties applied in this study. Results also show a dominating cold-rain process for all cases and a strong but mostly non-systematic impact on the release of latent heat, which is considered to be the prime mechanism for the upscale growth of small errors affecting the predictability of convective systems. The combined ensemble spread when accounting for all three uncertainties lies in the same range than the ones from an operational convective-scale ensemble prediction system with 20 members determined in previous studies. This indicates that the combination of different perturbations used in our study may be suitable for ensemble forecasting and that this method should be evaluated against other sources of uncertainty.

## 1 Introduction

Forecasting convective precipitation remains one of the key challenges in numerical weather prediction (NWP). Many aspects influence the predictability of convective precipitation, e.g., uncertainties in the synoptic-scale flow, inaccuracies in the state of the atmosphere and the underlying land surface, and approximations in the representation of key physical processes in numerical models. Although the processes triggering convection are broadly known, the ability to predict, in particular, severe

convective showers is still poor (Jorgensen and Weckwerth, 2003; Bennett et al., 2006; Barthlott and Hoose, 2015). Nowadays, most operational forecasting centers make use of convection-resolving ensemble modeling systems in which uncertainties in the initial and lateral boundary conditions as well as uncertainties in the representation of physical processes are accounted for (e.g., Clark et al., 2016; Barthlott and Barrett, 2020, and references therein). However, these ensemble modeling systems are often underdispersive (e.g. Bouttier et al., 2012; Raynaud and Bouttier, 2017) and the methodology for constructing such ensembles that effectively represent the numerous sources of uncertainty acting in nature remains an active field of research (Keil et al., 2019). To reduce the underdispersion of convective precipitation in convective-scale weather models, other sources of uncertainty need to be assessed. One candidate is the soil moisture content as it controls the partitioning of the available energy at the ground into sensible and latent heat. Land–atmosphere interactions are assumed to be decisive for cloud formation and subsequent convective precipitation and land-surface properties (e. g. land cover, terrain, and soil texture) are highly heterogeneous across a wide range of spatiotemporal scales. Therefore, it is difficult to establish potential relationships between land-surface variables and atmospheric variables such as temperature and precipitation. (e. g. Seneviratne et al., 2010; Schneider et al., 2018; Liu et al., 2022). Many studies documented the importance of soil moisture for convective precipitation and the complexity of the soil moisture–precipitation feedback which may vary spatially and temporarily (e.g., Pan et al., 1996; Findell and Eltahir, 2003; Hohenegger et al., 2009; Richard et al., 2011; Baur et al., 2018). As was documented by Hauck et al. (2011), soil moisture in models often shows a bias with respect to observations. The initial soil moisture content, however, is of great importance for precipitation forecasting: For drier soils, a systematic positive soil moisture–precipitation feedback was found by Barthlott and Kalthoff (2011), whereas for already relatively wet soils, the influence of increasing soil moisture is much weaker and not systematic anymore. In addition, horizontal land-surface wetness gradients can induce mesoscale circulations leading to convection initiation over dry soils (Taylor et al., 2012; Baur et al., 2018).

Besides the unclear role of the soil moisture–precipitation feedback, there are also large uncertainties arising from the microphysics of mixed-phase clouds. In current NWP models, aerosol–cloud interactions are considered one of the most uncertain processes (e.g. Tao et al., 2012; Altaratz et al., 2014; Fan et al., 2016; Barthlott and Hoose, 2018). In polluted environments, the activation of aerosol particles (serving as cloud condensation nuclei CCN) into cloud droplets results in more numerous and smaller cloud droplets. Known as the "lifetime effect", the onset of precipitation can be suppressed due to a weaker collision-coalescence process, which can result in a longer cloud lifetime (Albrecht, 1989). However, the effects of aerosols on convective precipitation have been shown to vary depending on cloud type, aerosol regime, and environmental conditions (e. g. Seifert and Beheng, 2006b; Khain et al., 2008; van den Heever et al., 2011; Tao et al., 2012; Barthlott et al., 2017). Moreover, there are large uncertainties in the aerosol number concentration because there exist only few in situ observations or routine measurements of aerosols in three-dimensional space (Thompson et al., 2021).

The cloud droplet size distribution (CDSD) is another source of uncertainty in convective precipitation simulations. The form of the underlying generalized gamma distribution is controlled by the shape parameter $\nu$, which determines the width of the size distribution and also the location of its maximum. Example size distributions illustrating the effect of different shape parameters will be given later in section 2.1. A higher shape parameter suppresses the autoconversion process of cloud droplets into raindrops, resulting in higher droplet number concentrations (e.g. Seifert and Beheng, 2001). The CDSD is also important

for the effective radius of cloud droplets, which is the relevant parameter for the radiative properties of clouds. However, the width of the cloud droplet size distribution is not well constrained by measurements and a wide range of values (between 0–14) based on cloud type and environmental conditions were reported (e.g. Levin, 1958; Gossard, 1994; Miles et al., 2000; Martins and Silva Dias, 2009). There are only few modeling studies on the effects of the shape parameter available, most are based on idealized simulations. For example, Igel and van den Heever (2017) have shown with large-eddy simulations of non-precipitating shallow cumulus clouds that changes were of the same order of magnitude as those due to a factor of 16 increase or decrease in aerosol concentration. In a recent work by Barthlott et al. (2022), the relative impact of varying CCN concentrations and different shape parameters of the CDSD were assessed for several convective cases over central Europe. They found a large systematic increase in total cloud water content with increasing CCN concentrations and narrower CDSDs together with a reduction in the total rain water content as a result of a less efficient collision-coalescence process. The precipitation response was generally larger for weakly-forced cases and averaged of Germany, the timing of convection was not sensitive to different CCN concentrations or shape parameters. Moreover, an increase in the shape parameter can produce almost as large a variation in precipitation as a CCN increase from maritime to polluted conditions. They also found that increasing CCN concentrations reduced the effective radius of cloud droplets more than larger shape parameters, but cloud optical depth had a similar increase with larger shape parameters as the change in aerosol loading from maritime to polluted. However, the impact of the shape parameter was assessed for one reference CCN concentration only and the need to determine the impacts of the shape parameter by combined sensitivity analyses was considered to be necessary. Furthermore, the shape parameter is one of the stochastically perturbed parameters in the widely used Thompson-Eidhammer cloud microphyics scheme and recent model results indicate a suitability of this parameter for generating ensembles at the convective scale (Griffin et al., 2020; Thompson et al., 2021).

While the individual perturbations of parameters or processes were conducted extensively in recent years, only few have investigated their combined effects. Imamovic et al. (2017) conducted convection-resolving simulations with a simplified land surface to dissect the isolated and combined impacts of soil moisture and orography on deep-convective precipitation for an initial profile corresponding to typical European summer climate conditions. They found a consistently positive soil moisture–precipitation feedback for horizontally uniform perturbations, irrespective of the presence of low orography. However, a negative feedback emerged with localized perturbations. Other studies with multiple-factor analyses exist mostly for idealized setups, e.g. for investigating the impact of environmental conditions and microphysics on the forecast uncertainty of deep convective clouds and hail using an emulator approach by Wellmann et al. (2020), for investigating aerosol–cloud–land surface interaction within tropical sea breeze convection (Grant and Heever, 2014) or investigating the relative sensitivity of a tropical deep convective storm to changes in environmental and cloud microphysical parameters (Posselt et al., 2019). Using the Morris one-at-a-time (MOAT) method for simultaneous perturbations of numerous parameters, Morales et al. (2019) explored the sensitivity of orographic precipitation within an environment of an atmospheric river. Schneider et al. (2019) investigated the relative impact of soil moisture and aerosols combined with orographic effects. They performed simulations with the COSMO (COnsortium for Small-scale MOdeling) model with 500 m grid-length for six real-case events over Germany with systematic changes in the initial soil moisture fields and different assumptions about the ambient aerosol concentration. The model produced a positive soil moisture–precipitation feedback for most of the cases with the soil moisture amount having

a stronger effect on precipitation than on its spatial distribution. The precipitation response to changes in the CCN concentration was found to be more complex and case dependent. However, both aerosols and soil moisture uncertainties were of similar importance for quantitative precipitation forecasting. Baur et al. (2022) studied the combined impact of soil moisture and microphysical perturbations with the COSMO model for a single case study of locally forced convection in central Europe. They found a large sensitivity of 12 h precipitation deviations ranging between $-23\%$ and $+10\%$ compared to a reference run.

In this study, we expand this line of investigation by perturbing the soil moisture, the CCN concentration, and the shape parameter simultaneously using a state-of-the-art operational numerical model. We choose these uncertainties because (i) their individual impact was documented in many recent studies and (ii) all have an impact on the life cycle of convection at different stages from its initiation to the decay. We will investigate the role of different aerosol amounts ranging from low CCN concentrations (representing maritime conditions) to very high CCN concentrations (representing continental polluted conditions) with different values of the shape parameter combined with a wet and dry soil moisture bias. By comparing the effect of ambient aerosol amount with changes in the shape parameter, we can quantify their relative impact on predicting convective precipitation under different soil moisture regimes. We further want to quantify the individual and collective effects of land-surface and microphysical uncertainties on convective-scale predictability for different weather regimes. The combination of different sensitivities will help answering the question if and how different processes compensate or enhance each other and how large the spread of different process pathways is. We can further answer the question if and how aerosol effects on clouds are modulated by soil moisture uncertainties (e.g. drier or wetter soils). The unique aspect of this work is that it is the first to systematically evaluate the collective effects of CCN concentrations and uncertainties in the CDSD for multiple cases with different synoptic controls and different initial soil moisture contents using a state-of-the-art operational numerical model.

## 2 Method

### 2.1 Model description and simulations overview

The model set-up is generally similar to that used in Barthlott et al. (2022), but is described here for reference. We use version 2.6.2.2 of the ICOsahedral Non-hydrostatic (ICON) model. ICON is a fully compressible model using an unstructured triangular grid with C-type staggering based on a successive refinement of a spherical icosahedron (Zängl, 2012; Zängl et al., 2015). It can be run in global and limited-area mode with grid-nesting capability. The convection-permitting configuration ICON-D2 at 2 km horizontal grid spacing is used at the German Weather Service (DWD) for operational forecasts over central Europe since February 2021. Model domain (Fig. 1), horizontal and vertical resolution used in this study correspond to the operational ICON-D2 configuration. Further model settings are presented in Table 1.

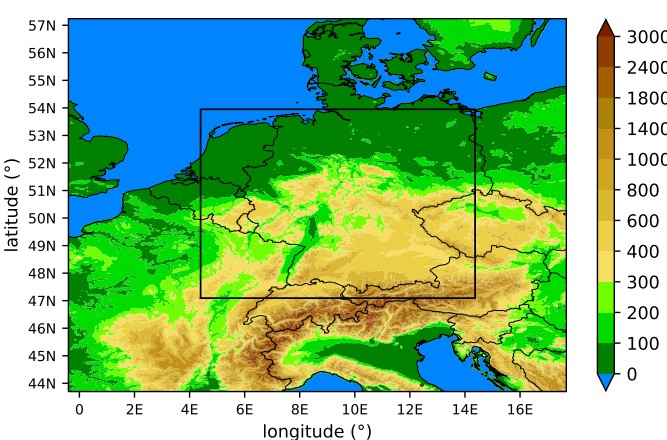

**Figure 1.** ICON simulation domain and model orography in meters above sea level. The black rectangle depicts the evaluation domain covering most of Germany and parts of neighboring countries.

**Table 1.** Model configuration for the ICON simulations.

| Model aspect | Setting |
| --- | --- |
| grid | unstructured triangular grid R19B07 (2 km grid spacing, 538164 cells) |
| vertical grid | vertically stretched smooth level vertical (SLEVE) coordinates (Leuenberger et al., 2010) |
| vertical levels | 65, 14 levels in the lowest 1000 m, lowest level at 10 m agl |
| model top | 23 km |
| initial and boundary data | 7 km ICON-EU analyses, 3 h update |
| time step | 20 s |
| initialization time | 00:00 UTC |
| integration time | 24 h |
| microphysics | double-moment bulk microphysics (Seifert and Beheng, 2006a) |
| heterogeneous ice nucleation | based on mineral dust concentrations (Hande et al., 2015) |
| homogeneous ice nucleation | following Kärcher and Lohmann (2002) and Kärcher et al. (2006) |
| convection parameterization | deep convection resolved explicitly |
| | Tiedtke-Bechtold scheme for shallow convection (Bechtold et al., 2008; Tiedtke, 1989) |
| land-surface model | multi-layer land-surface scheme TERRA (Heise et al., 2006) |
| turbulence parameterization | 1D based on prognostic equation for the turbulent kinetic energy (Raschendorfer, 2001) |
| radiation scheme | rapid radiation transfer model (RRTM, Mlawer et al., 1997), called every 12 min |
| data assimilation | none |

## CCN uncertainty

In contrast to the operational setup at DWD, we use the double-moment microphysics scheme of Seifert and Beheng (2006a) for representing aerosol effects on the microphysics of mixed-phase clouds. This scheme predicts mass and number concen-

tration of cloud water, rain water, ice, snow, graupel, hail and has been extensively used to study aerosol–cloud interactions in recent years with the ICON model (e.g. Heinze et al., 2017; Costa-Surós et al., 2020; Barthlott et al., 2022) and its predecessor, the COSMO model (e.g. Seifert et al., 2012; Barthlott et al., 2017; Barthlott and Hoose, 2018; Keil et al., 2019;

Marinescu et al., 2021). Pre-calculated activation ratios stored in look-up tables (Segal and Khain, 2006) are used to compute the activation of CCN from aerosol particles. The condensation nuclei are all assumed to be soluble and follow a bi-model size distribution (Seifert et al., 2012). Using the Segal and Khain (2006) activation, four different values of the number density of CNN ($N_{\text{CCN}}$) are available, representing maritime ($N_{\text{CCN}} = 100\,\text{cm}^{-3}$), intermediate ($N_{\text{CCN}} = 500\,\text{cm}^{-3}$), continental ($N_{\text{CCN}} = 1700\,\text{cm}^{-3}$), and continental polluted conditions ($N_{\text{CCN}} = 3200\,\text{cm}^{-3}$). Typical conditions of central Europe are represented

by the continental aerosol assumption (Hande et al., 2016).

**CDSD uncertainty**

The second perturbation consists of different widths of the cloud droplet size distribution (CDSD). The size distribution is based on a so-called generalized Gamma distribution as follows:

$$f(x) = Ax^\nu \exp(-\lambda x^\mu) \tag{1}$$

and depends on the shape parameter $\nu$ and dispersion parameter $\mu$ as a function of the particle mass $x$. With the predicted mass and number densities, both coefficients $A$ and $\lambda$ can be calculated (Seifert and Beheng, 2006a). A number of microphysical processes depend directly on the shape parameter (e.g. autoconversion, self collection) or indirectly (e.g. melting, evaporation, accretion, riming, sedimentation) leading to a potentially large impact of the CDSD on the simulated precipitation totals (Barthlott et al., 2022). In this study, we perturb the shape parameter from 0 to 8. These values lie in the observational range

and were shown to have a large impact on convective precipitation forecasts in recent studies (Barthlott et al., 2022; Baur et al., 2022; Matsunobu et al., 2022). The dispersion parameter is kept constant in all simulations ($\mu = 1/3$). To illustrate the impact of the shape parameter on the width of the CDSD, Fig. 2 shows example size distributions as a function of particle diameter $D$ at fixed cloud water content (QC) and cloud droplet number concentration (QNC) for different shape parameters. We refer to Khain et al. (2015) or Barthlott et al. (2022) for the conversion of Eq. 1 from particle mass $x$ to diameter $D$. It can be seen that

larger shape parameters narrow the size distribution and also shift the maximum to larger droplet sizes. An important feature is that with high shape parameters, the CDSD has less smaller droplets, but also less large droplets leading to a smaller effective radius, which impacts the optical properties of the clouds.

**Soil moisture uncertainty**

The third uncertainty included in our combined sensitivity analysis consists of three different soil moisture initializations.

Beside a reference run with initial values coming from the ICON-EU analysis, we conduct simulations with a dry and wet bias ($\pm 25\,\%$) to account for uncertainties in soil moisture. The value of 25 % was selected because Hauck et al. (2011) showed that simulated and observed soil moisture in southwestern Germany differ by around 20–30 %. Our procedure to include a soil moisture bias is as follows: At first, the soil moisture index (SMI) is converted to a volumetric water content (VWC) using the

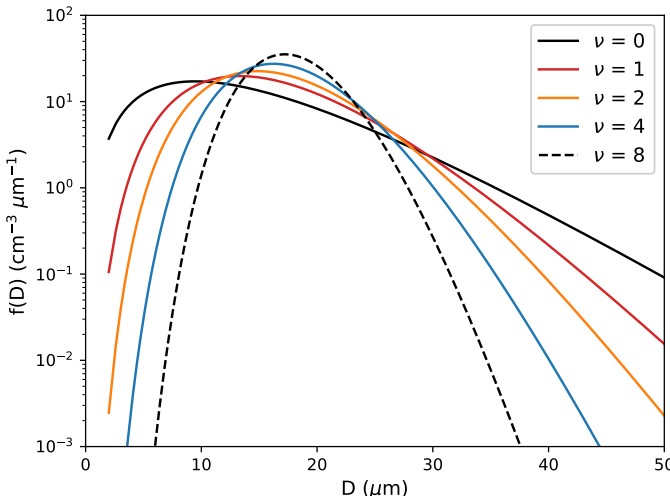

**Figure 2.** Cloud droplet size distributions for different values of the shape parameter $\nu$ at fixed cloud water content (QC = 1.0 g m$^{-3}$) and cloud droplet number concentration (QNC = 300 cm$^{-3}$). $D$ denotes the diameter of the droplets.

field capacity and the permanent wilting point for each soil type. Then the VWC is increased/decreased by a factor of $\pm 25\,\%$ at
every grid point in all levels. To assure physical meaningful values, we restrict the modifications to the limits of the air dryness point and the pore volume for each soil type. Finally, the modified SMI is calculated and written to the file used for model initialization.

In summary, this study investigates the effects of two microphysical uncertainties (i.e. CCN concentration and shape parameter) combined with land-surface uncertainties (soil moisture realizations). By applying four different CCN concentrations,
five different shape parameter values, and three soil moisture initializations, we end up with an ensemble of 60 model runs per case. The reference run uses unmodified soil moisture values, a continental CCN assumption typical for central Europe, and a shape parameter of 0. All model runs are abbreviated based on three letters:

1. soil moisture content: dry (DRY), reference (REF), or wet (WET)

2. CCN concentration: maritime (m), intermediate (i), continental (c), polluted (p)

3. shape parameter $\nu$: 0, 1, 2, 4, 8

The reference run would therefore be labeled as run REFc0. Including more microphysical uncertainties (e.g. ice nucleating particle concentration, hydrometeor sedimentation, or ice multiplication) as in idealized simulations by Wellmann et al. (2020) could be considered in the future, but were not performed at the moment due to the high number of possible combinations.

**Table 2.** List of cases with convective adjustment time scale $\tau$ and mean initial relative water content RWC for the three soil moisture scenarios.

| Synoptic-scale forcing | Date | $\tau$ (h) | RWC (DRY/REF/WET) (%) |
|---|---|---|---|
| weak | 5 June 2016 | 5.22 | 55/73/86 |
| weak | 9 June 2018 | 4.65 | 28/37/46 |
| strong | 10 June 2019 | 0.17 | 24/33/41 |
| strong | 17 August 2020 | 1.09 | 25/37/42 |

## 2.2 Case studies

We performed numerical simulations for a total of 4 days. To cover different typical weather regimes in central Europe, we selected two cases with weak and strong synoptic forcing, respectively (Tab. 2). Both weak forcing cases were also used in Barthlott et al. (2022), the strong forcing case of 17 August 2020 in Matsunobu et al. (2022). The 5 June 2016 case occurred during an exceptional sequence of severe thunderstorms in Germany, its meteorological situation is described in detail by Piper et al. (2016). Mohr et al. (2020) investigated the role of large-scale dynamics in an exceptional sequence of severe
thunderstorms of May–June 2018 in Europe to which the 9 June 2018 case belongs to. Large amounts of hail fell on 10 June 2019 where a severe storm system was passing over the city of Munich in southeastern Germany. This day was part of a 3 day storm series in June 2019 whose synoptic controls are given in Wilhelm et al. (2021).

To objectively quantify the degree of synoptic-scale forcing, we computed the convective adjustment time scale $\tau$ following Keil et al. (2014):

$$185 \quad \tau = 0.5 \left( \frac{\rho_0 c_p T_0}{L_v g} \right) \frac{CAPE}{P} \tag{2}$$

with reference values for density, $\rho_0 = 1.292\,\mathrm{kg\,m^{-3}}$ and temperature, $T_0 = 273.15\,\mathrm{K}$, specific heat of air at constant pressure $c_p$, latent heat of vaporization $L_v$, acceleration due to gravity $g$, convective available potential energy (CAPE), and the precipitation rate $P$. It is a measure to distinguish between different flow regimes and can be considered as an estimate of the time scale for the removal of conditional instability. Daily mean values of this time scale below a threshold of 3 h indicate strong
forcing, higher values weak forcing. A visual inspection of the synoptic weather charts in Fig. 3 confirms the results of the time scale analysis (Tab. 2).

We now briefly describe the synoptic situation of these cases and the 24 h rain distribution of the respective reference runs. Under weak synoptic forcing, there lies a dominating ridge in central Europe with a ridge axis over the Iberian Peninsula on 5 June 2016 and further to the east on 9 June 2018. On both days, low pressure systems are situated over the eastern Atlantic.
Over Germany, the surface pressure lies between 1012–1020 hPa with weak horizontal gradients and mid-tropospheric winds are weak from easterly (Fig. 3a) and southwesterly (Fig. 3b) directions. The 24 h accumulated precipitation of the reference runs shows scattered convective showers over Germany for these cases (Fig. 4a, b).

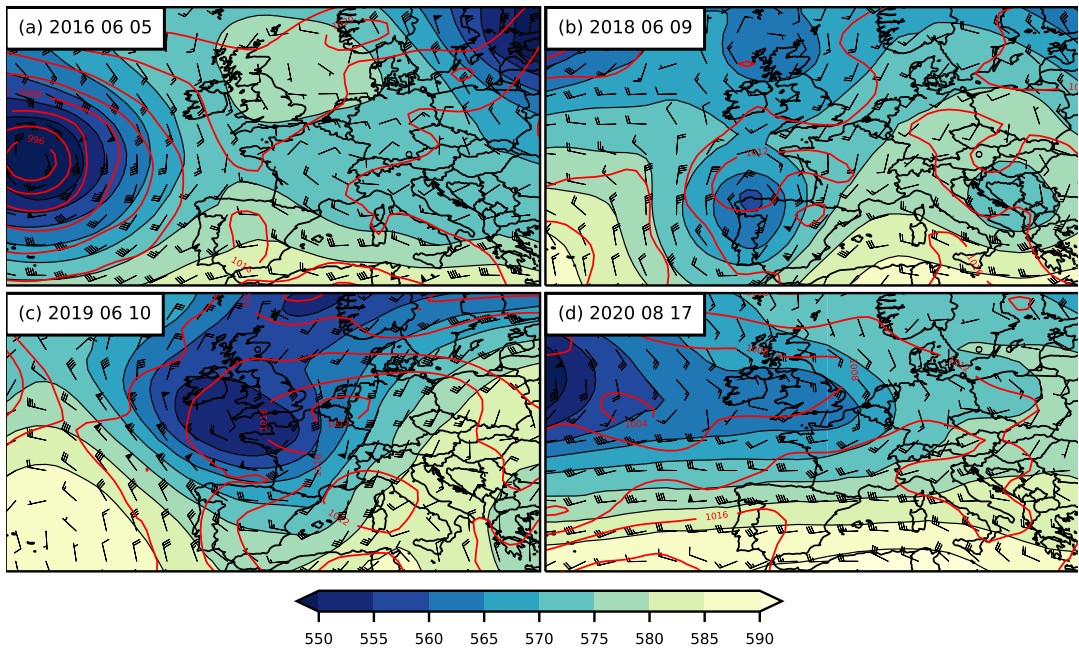

**Figure 3.** Global Forecast System (GFS) analyses at 12:00 UTC for the cases with weak **(a, b)** and strong **(c, d)** synoptic-scale forcing showing 500 hPa geopotential height (gpdm; shading), sea-level pressure (hPa, red contours), and 500 hPa wind barbs.

Under strong synoptic forcing, both analyzed days show a stronger baroclinicity of the flow (Fig. 3c, d) with a low pressure system northwest of France (10 June 2019) and over the eastern Atlantic (17 August 2020). Mid-tropospheric winds are more cyclonic from southwesterly directions with stronger winds on 10 June 2019 due to a deeper low and larger pressure gradients over Germany. The precipitation distribution reveals more organized convection and, especially on 10 June 2019, larger cloud clusters and more long-lived convection. We also compared the simulated precipitation to data from the precipitation analysis algorithm RADOLAN (Radar Online Adjustment) which combines weather radar data with hourly surface precipitation measurements of about 1300 automated rain gauges (not shown). For 24-h accumulated precipitation, we find an overall good agreement, even if the precise location of individual convective cells are not always simulated at the right place. However, the model succeeds reasonably well in reproducing the observed cloud and precipitation evolution which implies that these runs serve as a good basis for our combined perturbation experiments.

## 3 Results

### 3.1 Precipitation amount and timing

To investigate the response of precipitation amounts to the perturbations applied in this study, we computed domain-integrated precipitation totals for the evaluation domain given in Fig. 1. The percentage deviation for each of the 60 ensemble members

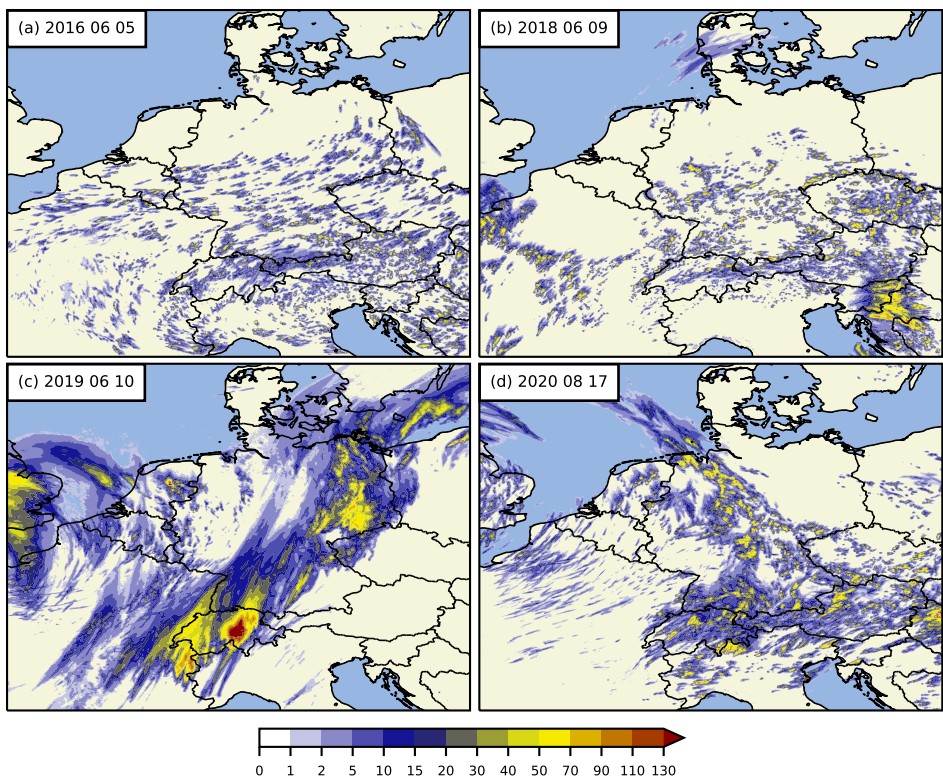

**Figure 4.** 24 h precipitation amount (mm) of the reference runs with continental CCN concentration, broad cloud droplet size distribution ($\nu = 0$), and reference initial soil moisture for the cases with weak **(a, b)** and strong **(c, d)** synoptic-scale forcing.

from the respective reference run are given in Fig. 5. We see a positive soil moisture–precipitation relationship for the vast majority of the performed simulations, as the accumulated precipitation in most runs with different shape parameters within one CCN concentration increases with increasing soil moisture independent of the type of synoptic forcing. Although not
shown here, the increase (decrease) of the initial soil moisture leads to a systematic increase (decrease) in CAPE and decrease (increase) in convection inhibition (CIN) for all cases independent of the microphysical uncertainties. As soil moisture controls the partitioning of the available energy at the ground into latent and sensible heat, the near-surface temperatures show a negative relationship to soil moisture, whereas specific humidity reveals a systematic positive relationship to soil moisture (not shown). This leads to an overall lower level of free convection resulting in larger amounts of CAPE with smaller CIN despite lower
boundary layer heights. However, the soil moisture impact is comparably weaker for the case of 5 June 2016. Especially for polluted conditions, there is only a marginal precipitation increase with soil moisture. This day, however, is characterized by already quite wet initial soils with an average relative water content of 73 % (see Tab. 2). As was pointed out by Barthlott and Kalthoff (2011), for already quite wet soils, the influence of increasing soil moisture is much weaker and the general response of precipitation to soil moisture is not systematic anymore. For drier soils, however, where evapotranspiration is controlled by
soil moisture, a systematic positive relationship of the 24 h accumulated precipitation to soil moisture exists, which can also

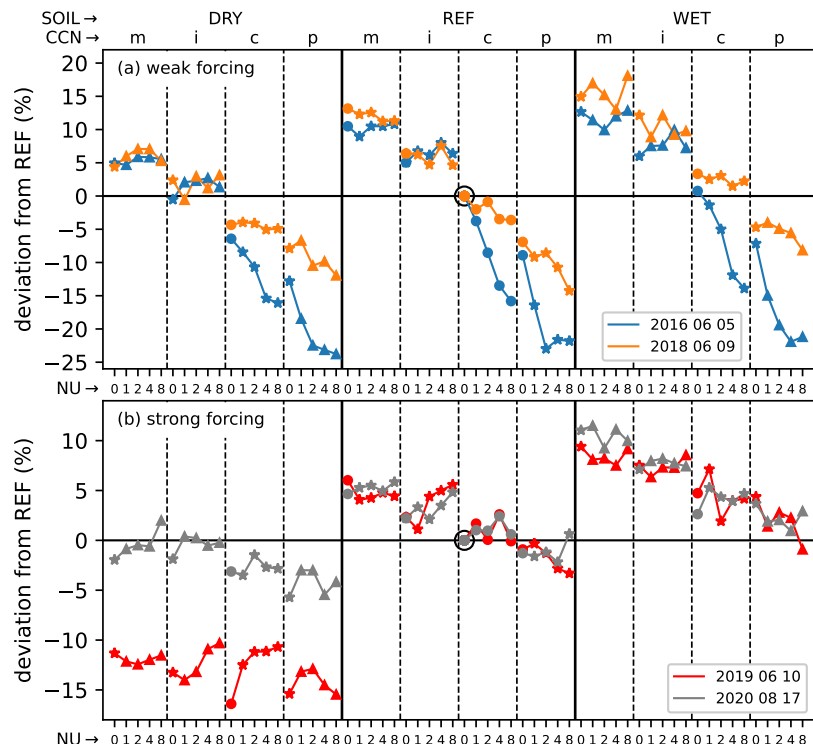

**Figure 5.** Precipitation deviation from the respective reference run (marked with a black circle) for **(a)** weak and **(b)** strong synoptic forcing. Data points are arranged in three blocks with different soil moisture contents (DRY, REF, WET), each of which is divided into four blocks with increasing CCN concentration (m, i, c, p). The points inside one CCN concentration then indicate the sensitivity with respect to the shape parameter (NU) from 0 to 8. The type of marking distinguishes between single effects (circles), double synergies (stars) and triple synergies (triangles).

be found in our simulations. The remaining days have considerably drier initial soil moisture values (36 % on 9 June 2018, 33 % on 10 June 2019, 34 % on 17 August 2020) and reveal a uniform positive soil moisture–precipitation feedback. There is one exception for the case of 9 June 2018: Only in the runs with a polluted atmosphere, more rain falls for three of the shape parameter runs in the dry run than in the reference run. On this day, the precipitation amounts in a polluted environment are

rather similar in the runs with dry and reference soil moisture. Reasons for that behavior could be related to the fact that a stronger thermal forcing with drier soils compensates the reduction of CAPE. It is worth noting that although stronger latent heat fluxes increase near-surface specific humidity, the impact on vertically integrated water vapor is negligible ($-1.4$ % to $+0.8$ %). This is important for the comparability of the runs from our 60-member ensemble. Although this study comprises only three different soil moisture realizations, we tested if there is a linear response of accumulated precipitation to the initial

soil moisture. Recently, a modeling study by Drager et al. (2022) suggests a new type of rainfall response to soil moisture in which intermediate-moisture soils receive less rainfall than do the driest or wettest soils. This non-monotonic soil moisture–precipitation relationship was found to result from the permanent wilting point's modulation of transpiration of water vapor by

plants. Our simulations revealed a monotonic soil moisture–precipitation relationship for all runs under strong synoptic forcing and for 85 % of the runs under weak synoptic forcing. Mean correlation coefficients were also high and ranged between 0.914 and 0.988. For more robust results, however, a higher number of soil moisture scenarios as applied in Dräger et al. (2022) or Barthlott and Kalthoff (2011) would be necessary.

The response of total precipitation to varying CCN concentrations shows a systematic precipitation decrease with increasing CCN concentrations for the weak forcing cases, irrespective of the underlying soil moisture content. This general trend is also apparent for the strong forcing cases, however, there are some shape parameter runs that deviate from this systematic behavior: e.g. for the runs at reference soil moisture, the intermediate CCN concentration with shape parameters between two and eight reveal larger rain amounts as with maritime CCN concentration for the 10 June 2019 case. The same applies to the case of 17 August 2020 at wet soils from continental to polluted conditions with a shape parameter of 0. However, the magnitude of the CCN response for the strong forcing cases is much lower ($-16.4$ % to $+11.5$ %) than for weak forcing ($-23$ % to $+18$ %) which is in agreement with previous findings regarding aerosol–cloud interactions with the COSMO model (Barthlott and Hoose, 2018; Keil et al., 2019) and with ICON (Barthlott et al., 2022). Note that different models may produce different responses to aerosol perturbations, but these studies used the same double-moment scheme for simulating convective episodes over central Europe. The validity of the convection invigoration mechanism proposed in Rosenfeld et al. (2008) is still open and many studies documented a decrease of total precipitation with increasing aerosol concentrations (e.g. Tao et al., 2012; Storer and van den Heever, 2013). Using idealized simulations, Grant and van den Heever (2015) showed that the influence of aerosols varies inversely with storm organization and Fan et al. (2009) found that vertical wind shear qualitatively determines whether aerosols suppress or enhance convective strength.

It is further of interest to analyze the impact of different shape parameters on precipitation deviations. For the weak forcing cases, there is only a small sensitivity of total precipitation for maritime and intermediate CCN concentrations. For more polluted environments (respective runs c and p), the sensitivity to the shape parameter becomes much larger and precipitation amounts tend to decrease with larger values of the shape parameter. The shape parameter impact for strong forcing cases remains generally small in all soil moisture and CCN concentration regimes. Furthermore, there is no systematic effect on precipitations totals observable. Only for runs DRYc0 to DRYc8 on 10 June 2019, there is a systematic precipitation increase with somewhat stronger precipitation deviations compared to the other cases.

A feature only apparent for weak synoptic forcing is the fact that for maritime and intermediate CCN concentration at dry soils, more precipitation is simulated as with reference soil moisture and continental conditions. The enhanced warm-rain process together with the stronger thermal forcing seems to balance the CAPE reduction for drier soils. For the strong forcing case of 17 August 2020, the maritime and intermediate runs with drier soils simulate similar precipitation amounts as the reference run. Obviously, the enhanced warm-rain process roughly balances the reduction in CAPE. To support this statement, we calculated the percentage deviations of CAPE and autoconversion/accretion of those model runs to the reference run. We find that the percentage magnitudes are almost identical: CAPE decreases by 11.8 % whereas the warm-rain process increases by 11.5 %. Although these variables cannot be used to quantitatively determine their impact on the total rain amount, it nevertheless supports our hypothesis that the CAPE reduction with drier soils can be compensated by the effects of a strengthened

warm-rain process. Interestingly, the reference runs in both cases with weak forcing still have larger precipitation amounts than those from the wet scenario with polluted CCN concentrations. This points towards a dominating precipitation reduction by a reduced collision-coalescence process over a soil moisture increase due to higher instability. An important result is that an increase in the shape parameter can cause almost as large a change in precipitation totals as a CCN increase from maritime to polluted conditions, and for weak forcing cases, the shape parameter has a larger effect on precipitation totals in polluted environments.

For a more complete picture of the rain response to our microphysical and land-surface perturbations, we also analyze the fraction of cloudy grid points of each model run. We find a mostly systematic increase in cloud cover with increasing CCN concentrations and larger shape parameters (not shown). Although the relative changes are considerable ($-13\,\%$ to $+29\,\%$), the absolute changes are much smaller and range between $+1\,\%$ to $+3\,\%$. Nevertheless, the increase in cloud cover seems contrary to the mostly decreasing total precipitation amounts with increasing CCN concentration and increasing shape parameter (see Fig. 5). The answer to this is twofold: (i) the reduced warm-rain process as a result of a less efficient collision–coalescence process leads to a longer cloud lifetime or (ii) stronger rain intensities must compensate the smaller cloud cover. We therefore now analyze the daily cycle of 30 min precipitation rates to investigate the reasons for the strong effects on precipitation totals and to assess if longer/shorter lifetimes or increased/reduced rain rates are simulated by the model.

The first case of 5 June 2016 is characterized by a similar diurnal cycle in all runs indicating that the initiation of convection is, on average, not sensitive to the perturbations applied in this study (Fig. 6). However, the maximum precipitation intensities are strongly modified ranging from $0.15\,\text{mm}$ ($30\,\text{min}^{-1}$) to $0.25\,\text{mm}$ ($30\,\text{min}^{-1}$). The higher the CCN concentration, the lower are the rain intensities. The range of the shape parameter runs within one CCN concentration is increasing with higher CCN concentrations. These features explain the lower precipitation amounts and the stronger precipitation deviation in polluted environments found in Fig. 5. The second weak forcing case (9 June 2018) generally shows a similar behavior. On this day, we also see a soil moisture impact: In the wet scenario, the increase in precipitation rates after 09:00 UTC is weaker than in the reference run. As the resulting maximum precipitation rates later on are mostly larger than with reference soil moisture and precipitation rates at the end of simulation time (21:00–00:00 UTC) are still higher as in the REF-run (probably due to the larger amounts of CAPE), precipitation totals are larger than in the REF-run.

The two strong forcing cases are characterized by larger mean rain intensities as the weak forcing cases as the ratio of rainy grid points is higher (see precipitation distribution in Fig. 4). As the timing is again broadly similar and the spread in rain intensities are generally smaller, the total sensitivity to our perturbations is therefore less than for weak forcing in agreement with the findings for the total precipitation deviations (Fig. 5). For both strong forcing cases, a systematic soil moisture impact is present: the higher the soil moisture, the higher are the maximum rain intensities which is most likely a direct effect of the CAPE increase. In addition, maximum precipitation rates in the dry runs for 10 June 2019 occur 1–2 h earlier as in the reference run. The earlier decaying phase of convection is then responsible for the strong total precipitation reduction on that day. The case of 17 August 2020 shows increasing precipitation rates with higher soil moisture, although the increase is less pronounced than on 10 June 2019. Again, largest precipitation rates are found for maritime CCN concentrations and the spread with respect to different shape parameters remains small.

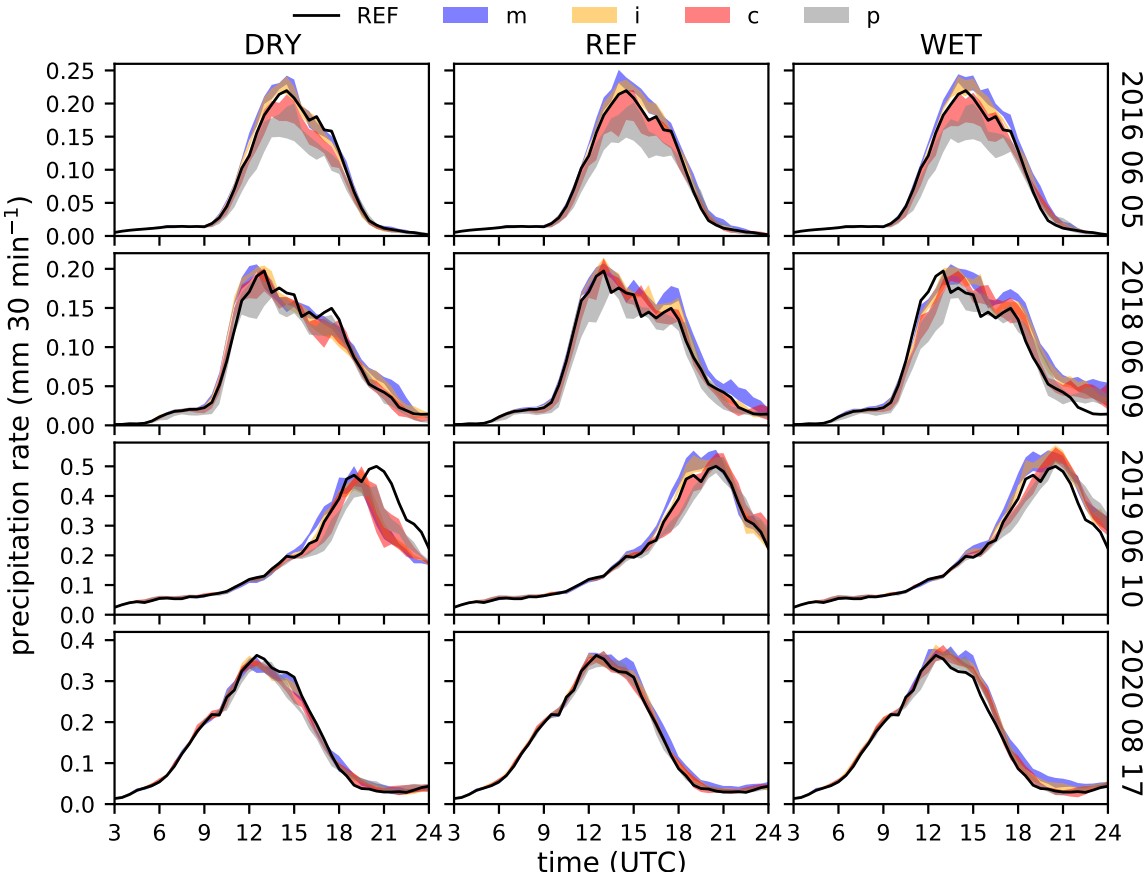

**Figure 6.** Domain-averaged precipitation rates for weak forcing (rows 1, 2) and strong forcing (rows 3, 4) for dry (left), reference (middle), and wet soil conditions (right). The color-coded areas indicate the range between the minimum and maximum precipitation rate for all shape parameter sensitivities within one CCN concentration. Different colors indicate the 4 different CCN concentrations. The black lines indicate the respective reference runs with reference soil moisture, continental CCN concentration, and a shape parameter of 0.

In order to quantify the ensemble spread, we computed the domain average of the grid point based normalized standard deviation $S_n$ as follows (see e.g. Keil et al., 2019):

$$S_n(x,y) = \frac{1}{\overline{P(x,y)}} \sqrt{\frac{1}{N-1} \sum_{i=1}^{N} \left\{ \overline{P(x,y)} - P_i(x,y) \right\}^2} \tag{3}$$

$P_i(x,y)$ denotes the hourly precipitation of member $i$, $\overline{P(x,y)}$ represents the ensemble mean hourly precipitation, and $N$ is the ensemble size. The normalization of the spread is done in order to remove fluctuations that are due to the diurnal variation of precipitation. Beside an ensemble spread based on all 60 members, we also computed the spread induced by soil moisture, CCN, and the shape parameter individually. As only three soil moisture regimes are available for each identical CCN concentration and shape parameter, we used the bootstrapping method to randomly pick between different suitable combinations

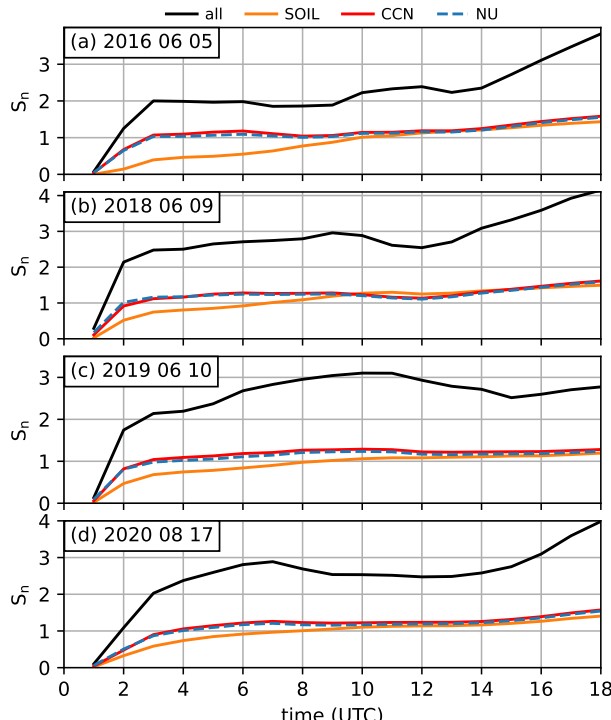

**Figure 7.** Domain-averaged normalized ensemble spread $S_n$ based on hourly precipitation amounts evaluated for different uncertainties (soil moisture SOIL, cloud condensation nuclei CCN, and shape parameter NU).

to calculate their normalised spread. This procedure was repeated 100 times. The results show that the area-averaged local precipitation variability introduced by varied CCN concentrations and shape parameters is rather similar (Fig. 7). This finding holds true for all days irrespective of the synoptic-scale forcing. For both uncertainties, the variability increases rapidly already in the first hours of the forecast, followed by a rather constant plateau until a further increase occurs in the afternoon

at the peak of convective activity (see rain rates in Fig. 6). In contrast to that, the variability due to soil moisture reveals a weaker increase early in the simulation and reaches similar high values (or even higher ones on 9 June 2018) as CCN and shape parameter variability only around noon. In the afternoon, a similar weak increase is simulated as in the other types of uncertainty. Later, soil moisture variability remains slightly below the ones from CCN and shape parameter. For a high-impact weather period of 2016, Keil et al. (2019) found that the spread induced by soil moisture was slightly larger than the one

induced from different CCN concentrations in the afternoon. However, in their study soil moisture was perturbed by applying high-, low- and bandpass filters to introduce surface perturbations which is different from our approach of using a soil moisture bias. Figure 7 further reveals that the overall ensemble spread when accounting for all three uncertainties during the active convective period lies between 2.2 and 4.1. More importantly, the absolute values lie in the same range than the ones from the operational convective-scale COSMO ensemble prediction system with 20 members at that time (Keil et al., 2019). This

indicates that the combination of different perturbations used in our study may be suitable to increase the variability and reduce the underdispersion of convective precipitation and that this method should be evaluated against other sources of uncertainty.

The spread of the model results and the impact of double and triple synergies was demonstrated so far with precipitation deviations, precipitation intensities, and the normalized ensemble spread. Although the quantitative interpretation of nonlinear interactions is not the main focus of this study, we used the factor separation methodology of Stein and Alpert (1993) in order to understand how aerosols, the shape parameter, and soil moisture may interact synergistically. This methodology is a simple way to show how multiple factors and their nonlinear interactions influence a predicted field and has been applied many times in atmospheric sciences, e.g. for aerosol-cloud-land surface interactions within tropical sea breeze convection (Grant and Heever, 2014) or for the effects of topography, convection, latent, and sensible heat fluxes on Alpine lee cyclogenesis (Alpert, 2011). For the four cases analyzed here, we find that the single impact of changing one parameter has a much weaker response as the double or triple synergies (not shown). Furthermore, all double synergies work to enhance accumulated precipitation, whereas all triple synergistic interactions reduce the precipitation in agreement with findings from Grant and Heever (2014). The triple synergies are greater than the double synergies by a factor of approximately three. Whereas the factor separation for the single impacts is always correlated to the rainfall difference compared to the respective reference run, the double and triple synergy terms rather represent the contributions of the synergistic interactions that occur. We must emphasize that synergy terms may not be meaningful for a field that has a finite range like total precipitation and when individual impacts of one parameter dominate the change of precipitation. Nevertheless, the results from the factor separation and ensemble spread shown before demonstrate the importance of considering synergistic effects for convective-scale predictability.

### 3.2 Hydrometeor contents and microphysical process rates

To understand the impact of our perturbations on the precipitation amount and timing, we now analyze vertically integrated hydrometeor contents (Fig. 8). We find a systematic increase in total cloud water with increasing CCN concentration and also with increasing shape parameters. This can be attributed to a reduced warm-rain process which is in agreement with previous studies (e.g. Tao et al., 2012; Storer and van den Heever, 2013; Barthlott and Hoose, 2018; Barthlott et al., 2022). We find a strong systematic reduction of autoconversion and accretion rates with larger aerosols and larger shape parameters (not shown). Especially for maritime CCN concentrations, the impact on the autoconversion process is substantial and ranges from a more than 800 % increase with low shape parameter to a 11 % increase for a shape parameter of 8 compared to the respective reference run. Interestingly, total cloud water contents are almost independent of the initial soil moisture. The spread of the runs with different shape parameters is larger for polluted environments. The impact of the microphysical uncertainties is substantial with reductions of more than 50 % and increases of more than 150 %.

For total rain water, we find an opposite behavior to that of cloud water, namely a systematic negative response to higher CCN concentrations and shape parameters. In contrast to cloud water, the spread of shape parameter runs is rather similar in all CCN regimes. The overall response is smaller than for cloud water, but still large and ranges between $-55$ % to $+58$ % compared to the reference run. There is also a very weak sensitivity to soil moisture, but with positive and negative relationships depending on the day and specific configuration. An important finding is the fact that the percentage range is rather similar for

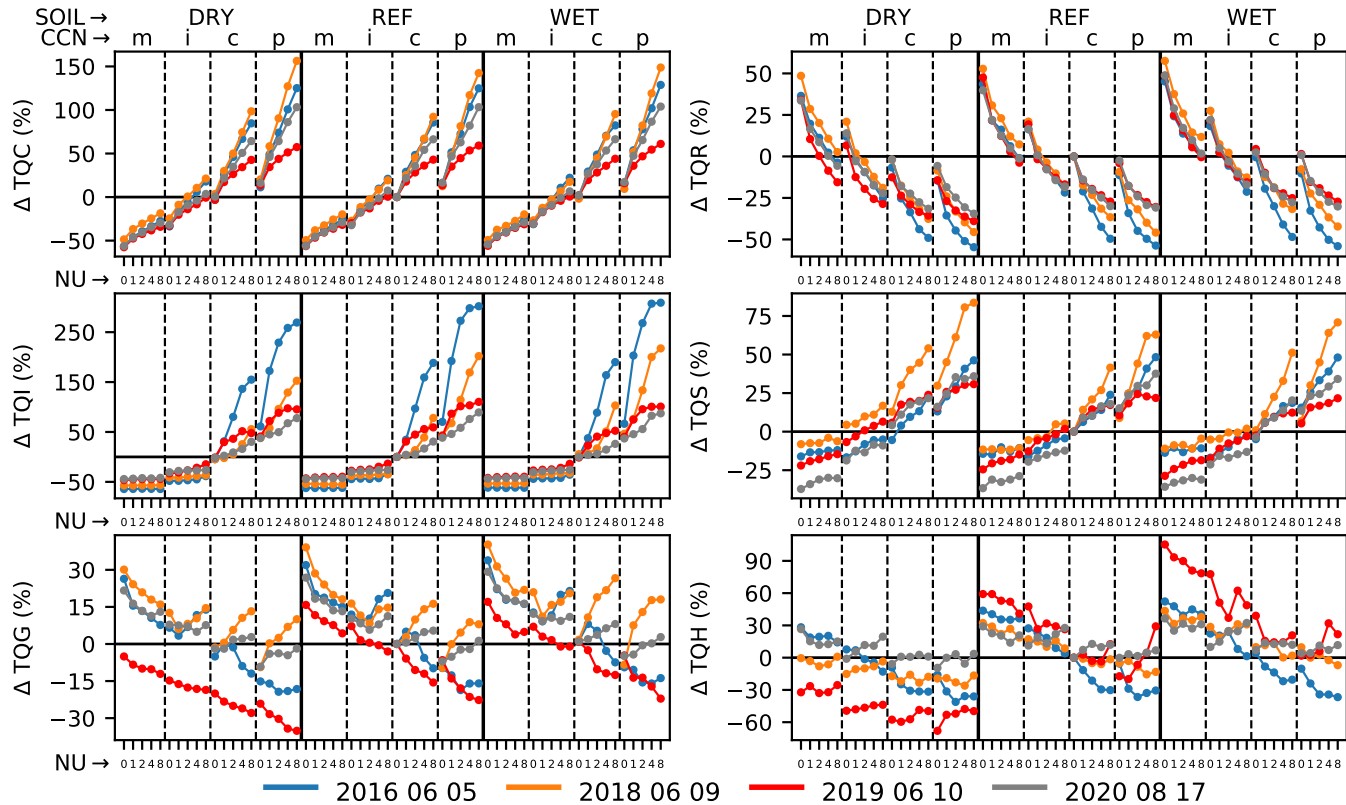

**Figure 8.** Percentage deviations from the respective reference runs of spatiotemporal averages of total cloud water (TQC), rain water (TQR), ice (TQI), snow (TQS), graupel (TQG), and hail (TQH). Data points are arranged as in Fig. 5.

different CCN concentrations and different shape parameters, indicating that the CCN concentration and the shape parameter

are equally important for total rain water deviations. For both cloud and rain water, show the weak forcing cases a stronger sensitivity than the ones with strong synoptic forcing.

The total ice content is increasing with higher CCN concentrations, but shows a remarkable feature: For rather clean environments (i.e. maritime and intermediate runs), the total ice content is not sensitive to different shape parameters and remains constant in all runs with the respective CCN concentrations. Only the intermediate CCN concentration with large shape pa-

rameters of $\nu = 8$ shows a weak increase in total ice. With larger aerosol amounts, there is also a strong dependance of the ice content to the shape parameter. The strongest increase is simulated for polluted environments with an increase of $+66\,\%$ to $+309\,\%$ on 5 June 2016 compared to the reference run. The positive relationship to the aerosol content was also found by Barthlott et al. (2022) and is probably related to the larger water load at higher levels caused by the reduced warm-rain process. The independance of the ice content to varied shape parameters in cleaner environments can be explained by the response of

heterogeneous and homogeneous freezing rates to our perturbations. We find that in all cases, there is almost no dependance of homogeneous freezing with varied shape parameters whereas at continental and polluted conditions, a strong sensitivity

exists (not shown). Heterogeneous freezing, although being mostly much smaller than homogeneous freezing, only reveals a small sensitivity to different shape parameters in clean environments. We therefore conclude that smaller amounts of super-cooled liquid water with a broad size distribution in cleaner environments is less susceptible to the impact of narrowing the size distribution with varied shape parameters.

The total snow content is generally increasing with larger aerosol loads and also shows a weak increase to larger shape parameters at maritime and intermediate CCN concentrations. The spread of the different shape parameter runs is increasing with CCN concentrations. The overall response is smaller than the one from ice, i.e. $-37\,\%$ to $+84\,\%$. Both the ice and the snow content are rather independent of the initial soil moisture assumption.

A more complex response is simulated for the total graupel content. For 10 June 2019, the model simulates an increase with soil moisture, an increase with CCN concentration, but a mostly systematic decrease with increasing shape parameters. The remaining three cases show a weak response to soil moisture only and especially at higher CCN concentrations also a increase with larger shape parameters. Similar results were found by Barthlott et al. (2022) where some of the days show a graupel increase and others a decrease with narrower CDSD. Comparing the absolute graupel contents of the cases analyzed reveals that the case of 10 June 2019 has a much larger graupel content than the remaining three cases. Some of the cases show decrease in graupel mass for maritime CCN conditions and an increase for higher CCN concentrations. This can be attributed to graupel/hail riming with cloud droplets which increases with larger shape parameters for already more narrow size distributions (not shown). Overall, the total graupel content exhibits a smaller response than snow ($-35\,\%$ to $+40\,\%$). The total hail content shows a strong response to soil moisture for the case of 10 June 2019 and a weaker, but still positive response for the remaining three cases. The CCN impact is mostly negative, only some runs with larger shape parameters reveal an increase at polluted environments. The impact of larger shape parameters is weaker than the one from CCN and is mostly negative or neutral.

It is now of interest to study the contribution of individual microphysical processes to the production and loss of rain water and to analyze the sensitivity and the magnitude to the perturbations applied in this paper. We therefore computed a rain water budget (B) which consists of the sources autoconversion (AC), accretion (ACC), melting (MELT) and sinks from evaporation (EVAP), riming (RIM), rain freeze (FR) as follows:

$$B = AC + ACC + MELT - EVAP - RIM - FR \tag{4}$$

We apply Eq. 4 to domain averages of vertical integrated and time accumulated process rates (Fig. 9). It can be seen that the melting of frozen hydrometeors is dominating the contribution to rain water production and always has a contribution of more than $50\,\%$. For the strong forcing case of 10 June 2019, its contribution is even larger (over $70\,\%$) than on the other days as the total absolute solid hydrometeor contents (ice, snow, graupel, hail) are the largest of all cases (not shown). Accretion possesses the second largest contribution for rain formation. Melting and accretion reveal an opposing response to increased shape parameters: The contribution of melting increases with larger shape parameters whereas the one from accretion is decreasing. This highlights the greater importance of cold-rain processes when the warm-rain process is reduced. The contribution of autoconversion is minor in all cases. Only with maritime CCN concentrations, there is a larger contribution of autoconversion to rain water formation from $9\,\%$ to $13\,\%$ for low shape parameters (broad CDSD). With higher shape parameters, its contribution

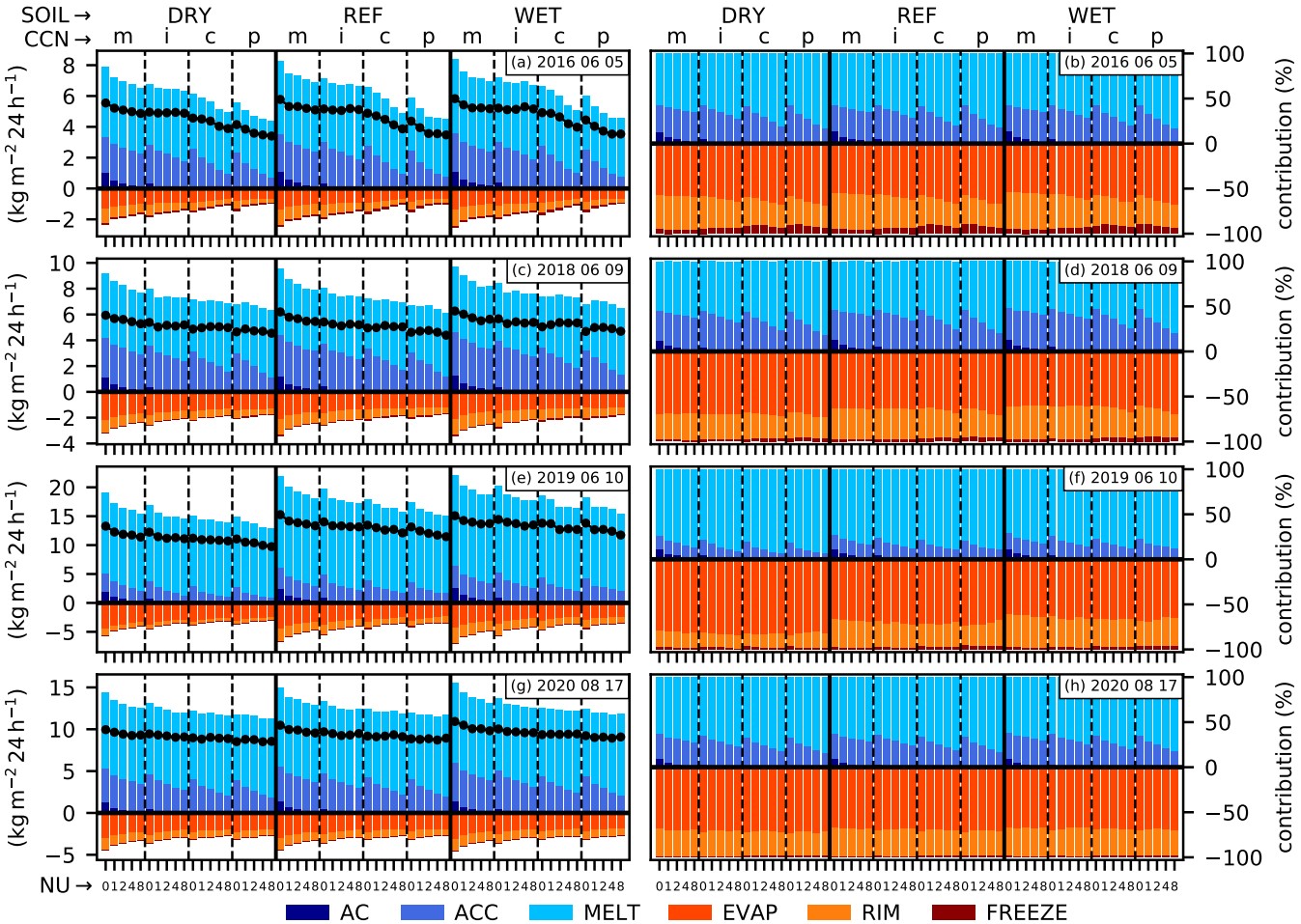

**Figure 9.** Rain water budget **(a, c, e, g)** with sources from autoconversion (AC), accretion (ACC), melting (MELT) and sinks from evaporation (EVAP), riming (RIM), rain freeze (FR) with black points indicating the overall budget (i.e. sources minus sinks) and percentage contributions **(b, d, f, h)** of individual process rates to sources and sinks, respectively. Bars are arranged as the data points in Fig. 5.

rapidly decreases to 3 % to 5 %. For the remaining higher CCN concentrations, the contribution of autoconversion is negligible. The absolute values of the sum of source terms decrease with increasing CCN concentrations as well as with increasing shape parameters. The relative change is higher for CCN differences as for different shape parameters.

The dominant sink term in our budget analysis is rain evaporation, followed by riming. The freezing of rain is below 5 % for the both strong forcing cases and on 9 June 2018, only on 5 June 2016, the contribution is larger (4–11 %). The response of the relative contributions from evaporation and riming to CCNs and shape parameters is rather weak and non-systematic. The absolute values of riming decrease in a systematic way with increasing CCN concentrations, but show no systematic reaction to variations in the shape parameter. The response of raindrop evaporation reveals a feature already found in earlier work (e.g. Barthlott et al., 2022): Total evaporation rates are highest for clean environments and are systematically decaying with both

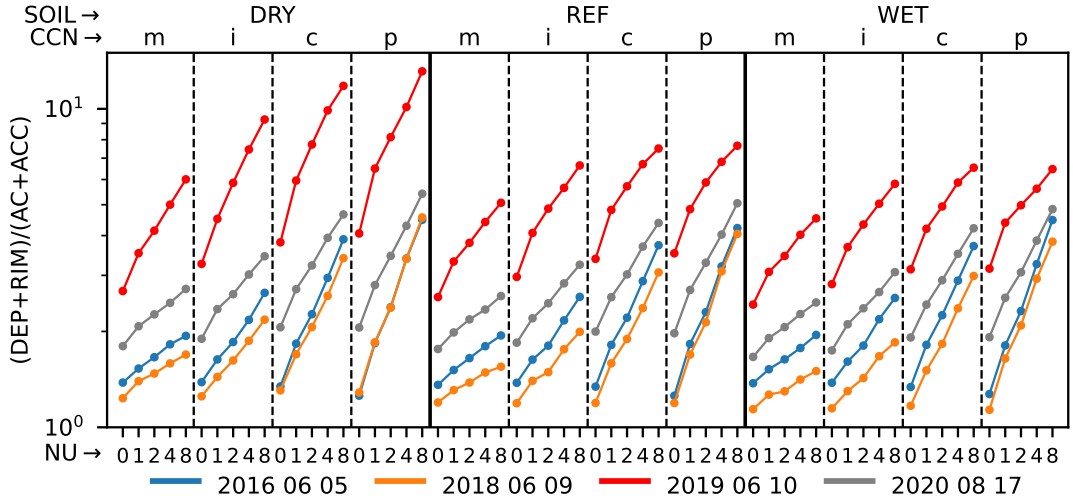

**Figure 10.** Ratio of cold-rain formation (deposition DEP and riming RIM) to warm-rain formation (autoconversion AC and accretion ACC). Data points are arranged as in Fig. 5.

increasing CCN and larger shape parameters. The combined effect of high aerosol loads with more narrow size distributions by larger shape parameters can lead to a evaporation increase of 57 % and a 30 % reduction compared to the reference simulation. This large effect can be related to the raindrop size distribution. As the aerosol concentration and shape parameter are increased, the size distribution shifts to populations of raindrops that are fewer in number but larger in size and evaporation is reduced due to the smaller surface area of large raindrops relative to their volume (see e.g. Storer et al., 2010; Barthlott et al., 2017). The sum of all source terms is decreasing with increasing CCN concentrations and larger shape parameters. But as the sink terms also reveal a reduction in magnitude, the net rain water budget is reduced less than expected from the reduced source terms. We also see a weak positive soil moisture impact which was already apparent in the analysis of the surface rain amounts (Fig. 5).

It is also of interest to assess the relative role of warm-rain and cold-rain processes and how they depend on the microphysical and land-surface uncertainties applied in this study. Therefore, we inspect the ratio of cold-rain formation (vapor deposition and riming) to warm-rain formation (autoconversion and accretion) integrated over the entire simulation time (Fig. 10). Our simulations show that on average, the cold-rain contribution is dominating the rain formation for all days. In agreement with the reduction of autoconversion and accretion documented earlier, the relative role of the processes via the ice phase is increasing with larger shape parameters and increased CCN concentrations. It is also evident that the higher the CCN concentration, the larger is the response to varied shape parameters. For example on 6 June 2018, the shape parameter induced increase in maritime environment is from a ratio of 1.38 to 1.94 (increase factor 1.41), whereas in the polluted environment, the increase is from 1.26 to 4.48 (increase factor 3.6) in the runs with drier soil moisture. The higher shape parameter sensitivity in polluted conditions was also seen, e.g., for total hydrometeor contents (see Fig. 8). As expected, the case with the highest ice water path (10 June 2019) shows the largest ratio of cold-rain to warm-rain formation of all analyzed cases. The ratios found with our

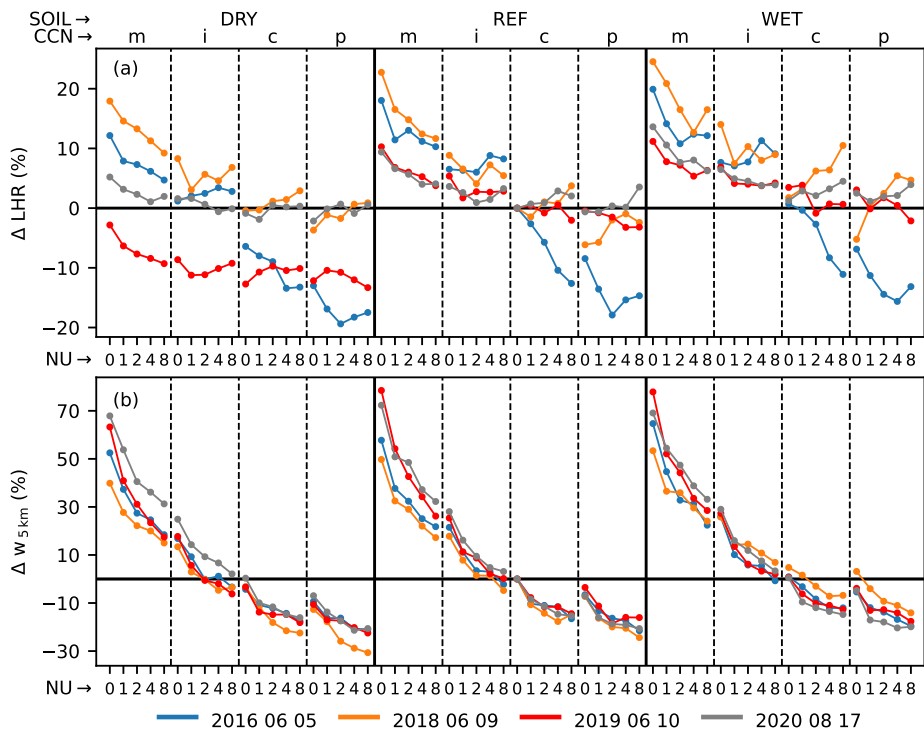

**Figure 11.** Percentage deviations of latent heat release (LHR) **(b)**, and deviations of updraft velocities $w$ at 5 km amsl. The latter were calculated for cloudy grid points only, for which vertically integrated total cloud water content is larger than $0.3\,\mathrm{kg\,m^{-2}}$. Data points are arranged as in Fig. 5.

ICON simulations lie in a similar range as in previous COSMO simulations using the same two-moment microphyiscs scheme (Barthlott and Hoose, 2018; Baur et al., 2022).

## 3.3 Impact on latent heat release and updraft velocities

Microphysical processes have an impact on atmospheric dynamics, in particular via the release of latent heat during condensation, deposition, riming, or freezing. Latent heat release in regions of precipitation is considered to be the prime mechanism for the upscale growth of small errors affecting the predictability of convective systems (e.g. Selz and Craig, 2015). Whereas the relative role of cold-rain processes increase with more narrow size distributions (see Fig. 10), the absolute values are declining (see budget analysis in section 3.2). This is the reason why the release of latent heat shows an opposing response to the ratio of cold-rain to warm-rain formation (Fig. 11a). For maritime CCN concentrations, the model simulates a mostly systematic decreasing latent heat release. There is not much of a sensitivity of the latent heat release to different shape parameters in the intermediate CCN regime and for larger aerosol loads (continental, polluted), the shape parameter sensitivity is case dependent, whereby either an increase or decrease is simulated. Compared to the respective reference run, the magnitude of latent heat

response is quite large and ranges between $-19.4\%$ to $+24.5\%$. There is also a weak positive soil moisture response for most cases, with again the strong forcing case of 10 June 2019 showing the strongest impact.

To study the impact on the dynamics, we computed spatiotemporal averages of updraft velocities (i.e. only positive values were accounted for) for cloudy grid points defined with a total cloud water content larger than $0.3\,\mathrm{kg\,m}^{-2}$ (Fig. 11b). For this analysis we selected the vertical updrafts at a height of $5\,\mathrm{km}$ agl because mean profiles reveal a systematic response of updraft velocities throughout the entire troposphere and the maximum differences between different aerosol loads occur between 5 and $6\,\mathrm{km}$ agl. The largest updraft velocities occur for maritime CCN concentrations and low shape parameters. The mean updraft speeds then decline with larger aerosol loads. For almost all analyzed cases, the updraft strength also declines with larger shape parameters. The impact is larger in clean environments, irrespective of the initial soil moisture assumption. Compared to the reference simulation, the differences are substantial ($+78\%$ to $-31\%$). The decline of updraft strength mostly follows the decline of latent heating, only at continental and polluted CCN concentrations, there are some cases which are not correlated well. However, the overall correlation of all data points within one soil moisture regime is still very high and ranges between 0.81 and 0.9 (not shown). As in previous studies with the ICON model (Barthlott et al., 2022), there is no CCN-induced convective invigoration as the updraft strength always declines with increasing CCN concentrations. In contrast to the theory suggested by Rosenfeld et al. (2008), the higher number concentrations of cloud droplets and the larger water load in the mixed-phase region do no increase latent heating in our simulations. Thus, updrafts are less buoyant and convection is not invigorated. However, the validity of the invigoration hypothesis remains open as there are contradictory results depending on the details of the microphysics scheme, the environmental conditions, or the cloud type (e.g. Altaratz et al., 2014; van den Heever et al., 2011; Fan et al., 2017). A recent study of Igel and van den Heever (2021) used theoretical calculations to demonstrate that a CCN-induced increase in storm updraft velocity is theoretically possible, but much smaller (and often negative) than earlier calculations suggested.

## 4 Summary and conclusions

The purpose of this study was to quantify the individual and collective effects of land-surface and microphysical uncertainties and to assess their impacts on convective-scale predictability. We therefore constructed a 60-member ensemble with the ICON model for four cases with different synoptic-scale forcing. The ensemble consists of combined perturbations of varied aerosol concentrations (maritime to polluted), different shape parameters of the cloud droplet size distribution (0–8), and different initial soil moisture conditions (dry and wet bias). We find a systematic positive soil moisture–precipitation feedback for all cases independent of the type of synoptic forcing as a result of higher instability. The precipitation response to different CCN concentrations and the shape parameter is stronger for weak forcing than for strong forcing in agreement with previous studies (e.g. Schneider et al., 2019; Keil et al., 2019; Barthlott et al., 2022). While the days with weak forcing show a systematic decrease in precipitation as the aerosol load increases, days with strong forcing also show non-systematic responses for some shape parameter values. For weak forcing, the response to the shape parameter is small for maritime and intermediate CCN concentrations, but becomes much larger in polluted environments. On the other hand, the shape parameter impact remains

generally small in all soil moisture and CCN concentration regimes for the strong forcing cases and no systematic effect on precipitations totals can be identified. Based on the magnitude of the total precipitation response ranging between $-16.4\,\%$ to $+11.5\,\%$ for strong forcing and ($-23\,\%$ to $+18\,\%$) for weak forcing, we conclude that the uncertainties investigated here and, in particular, their collective effects are highly relevant for quantitative precipitation forecasting of summertime convection in central Europe. These values gain even more importance if we consider that they are mean values over a large area and that the location and intensity of precipitation certainly varies locally with further implications for hazard assessments of convective storms. In some cases, we also find compensating effects of different disturbances. For example, the enhanced warm-rain process in clean environments can offset the CAPE reduction for drier soils and lead to an increase in precipitation compared to the reference run.

The analysis of the cloud fraction and rain intensities revealed that the overall timing of convection is not sensitive to the microphysical perturbations applied in this study. From this it can be concluded that different rain amounts are only caused by stronger or weaker rainfall intensities. The maximum rain intensities are higher for clean environments with a broad drop size distribution. An important finding is the fact that the combined ensemble spread when accounting for all three uncertainties lies in the same range than the ones from the operational convective-scale COSMO ensemble prediction system with 20 members determined in previous studies (Keil et al., 2019). Similarly, the factor separation methodology showed a larger impact of triple synergies to the simulation results compared to double synergies or single impacts which demonstrates the importance of considering synergistic effects for convective-scale predictability. To our knowledge, only Grant and Heever (2014), Park and van den Heever (2021) and Baur et al. (2022) studied synergistic effects of aerosols and soil moisture so far. Grant and Heever (2014) conducted idealized cloud-resolving simulations of tropical sea breeze convection and found precipitation reductions by over 40% and 50% for the most extreme perturbations. Park and van den Heever (2021) have performed two large idealized 130-member ensembles that represent different initial conditions typical of tropical sea breeze environments in which they simultaneously perturbed six atmospheric and four surface parameters. Comparisons of the clean and polluted ensembles demonstrated that aerosol direct effects reduce the incoming shortwave radiation, as well as the outgoing longwave radiation, within the polluted ensemble and that enhanced aerosol loading results in a weakening of the convection initiated along the sea breeze front. The realistic convection-resolving simulations of Baur et al. (2022) were conducted for a single case study only, but they found a similar sensitivity of precipitation deviations as in this study ($-23\,\%$ and $+10\,\%$).

The analysis of vertically integrated hydrometeor contents shows a strong systematic increase in total cloud water content with increasing CCN concentrations and larger shape parameters along with a decrease in total rainwater content. This could be attributed to a systematic decrease in the warm-rain processes of autoconversion and accretion. The impact of the microphysical uncertainties is substantial with variations in total cloud water ranging between $-50\,\%$ and $+150\,\%$ and in rain water between $-55\,\%$ to $+58\,\%$ compared to the reference run. Interestingly, cloud ice is insensitive to shape parameter variations in clean environments, whereas the CCN-induced ice increase is fostered with larger shape parameters in polluted environments. It is hypothesized that the smaller amounts of supercooled liquid water with a broad size distribution in cleaner environments are less susceptible to the impact of narrowing the size distribution with the shape parameter. The results from a rain water budget analysis revealed that melting of frozen hydrometeors is dominating the rain production, followed by accretion and an

only minor contribution from autoconversion. Whereas the contribution of melting increases with larger shape parameters, the one from accretion decreases. This opposing response highlights the greater importance of the cold-rain processes when the warm-rain process is reduced. The evaporation of raindrops proved to be by far the most important sink term with the largest values occurring in clean environments with broad size distributions. Larger aerosol loads and/or higher shape parameters lead to raindrop distributions that are larger in size and therefore less susceptible for evaporation. The sum of all source terms decreases with increasing CCN concentrations and larger shape parameters. The relative change is larger for CCN differences than for various shape parameters. However, because the sink terms also show a decrease in magnitude, the net rainfall budget is reduced less than expected due to the reduced source terms.

Finally, our results also showed a dominant cold-rain process for all cases and a stronger relative role of processes via the ice phase at larger shape parameters and increased CCN concentrations. Compared to the respective reference run, the magnitude of latent heat response is quite large and ranges between $-19.4\%$ to $+24.5\%$. Consistent with previous work with the ICON model (Barthlott et al., 2022), there is no CCN-induced convection invigoration with updrafts being less buoyant when the CCN concentration is increased.

Our findings demonstrate that aerosols and the shape parameter of the CDSD are both important for quantitative precipitation forecasting. Especially, the concept of combined perturbations based on realistic parameter perturbations in combination with soil moisture heterogeneities can provide a good ensemble spread. This indicates that the combination of different perturbations used in our study may be suitable for ensemble forecasting and that this method should be evaluated against other sources of uncertainty. First efforts in this direction have been performed by Matsunobu et al. (2022) who investigated the relative importance of microphysical uncertainties on cloud and precipitation forecasts implemented in a ICON-D2 ensemble prediction system on different spatial and temporal scales for five real cases in central Europe. Given the overall large impact of uncertainties due to aerosols and the shape parameter identified in this work, the use of a stochastically perturbed parameter (SPP) scheme for these microphysical uncertainties could be beneficial and should be pursued in future work.

*Data availability.* ICON model output is available on request from the authors.

*Author contributions.* CB and CK developed the project idea and designed the numerical experiments. AZ performed the numerical simulations. CB and AZ conducted the analyses, TM calculated the convective adjustment time scale, and all contributed to the interpretation of the results. CB wrote the paper, with contributions from all co-authors.

*Competing interests.* The authors declare that they have no conflict of interest.

*Acknowledgements.* The research leading to these results has been done within the subproject B3 of the Transregional Collaborative Research Center SFB/TRR 165 "Waves to Weather" (www.wavestoweather.de) funded by the German Research Foundation (DFG). The authors wish to thank the German Weather Service (DWD) for providing the ICON model code and the initial and boundary data. This work was performed on the supercomputer ForHLR funded by the Ministry of Science, Research and the Arts Baden-Württemberg and by the Federal Ministry of Education and Research.

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
