# Peer review of "Impacts of combined microphysical and land-surface uncertainties on convective clouds and precipitation in different weather regimes"

_Atmospheric Chemistry and Physics, 2022_

## Author Comment (AC1)

**Responses to the reviewers**

Impacts of combined microphysical and land-surface uncertainties on convective clouds and precipitation

by C. Barthlott, A. Zarboo, T. Matsunobu, and C. Keil                    August 1, 2022
* * *
We thank both reviewers for reading the manuscript and providing detailed comments. We have carefully considered all comments and changed the manuscript accordingly. Please find below our responses in blue.

**Reviewer 1**

This paper examined the precipitation and updraft changes over Germany and parts of neighboring countries to four different parameters: CCN concentration, shape parameter, soil moisture, and synoptic forcing.

The first result section is a series of descriptions of what's shown in the simulations rather than a coherent synthesis/comparison. The reviewer strongly suggests that the authors develop a better way of organizing and displaying the results, particularly for the precipitation responses to three perturbed parameters. The authors need to separate materials into different subsections focusing on one perturbed parameter at a time and demonstrating the synergistic effects explicitly. Furthermore, the results are discussed when one of four parameters is constrained. For example, Figure 6 and associated texts illustrate the impact of soil moisture under different synoptic setups. As such, examination of the synergistic/interactive effects of more than two perturbed parameters is absent, which is critical in terms of adding novel findings to the existing literature on modulation of convective responses by interactions between aerosol and soil moisture (or other environmental parameters). Addressing the synergistic is imperative since the title is Impacts of "combined" microphysical and land-surface uncertainties. Also, since the simulation results are quite different depending on the synoptic setting, the reviewer recommends authors to revise the title to include the synoptic aspect.

We appreciate the suggestion from the reviewer but believe that a separation into different subsections focusing on one perturbed parameter alone would rather complicate the demonstration of the synergistic effects. We also believe that the synergistic effects are visible from the precipitation deviations (Fig. 5) and precipitation rates (Fig. 6), because all realizations of the ensemble are visualized here and the synergetic effect of changing 3 parameters can be identified. In order to better differentiate between the individual and combined impacts, we changed the marker type in Fig. 5 to distinguish between single effects (circles), double synergies (stars) and triple synergies (triangles). Reviewer 2 recommended to include the individual contribution to the spread from the 3 different parameters and we included a new Figure 7 for that. We believe that this new content is related to this reviewer's comment as well. Based on the last review comment (check Grant and van den Heever (2014) for how they computed synergistic effects), we applied the factor separation methodology of Stein and Alpert (1993) to our simulation results and give an example here (Fig. R.1). As already mentioned in the article of Grant and van den Heever (2014), the interpretation of the results obtained from the factor separation calculations, particularly for a field that has a finite range like total precipitation, is not always trivial. We find that all double synergies work to enhance accumulated precipitation, whereas all triple synergistic interactions reduce the precipitation in agreement with findings from Grant and van den Heever (2014) who studied the impacts of soil moisture, CCN concentrations, and roughness length on tropical sea breeze convection. The magnitudes of all double and all triple synergies are quite similar, respectively. The single impacts displayed in blue are much smaller and are correlated

[Figure]

Figure R.1: Factor separation analysis of single impacts (blue), double synergies (orange), and triple synergies (red) for the case of 05 June 2016.

to the precipitation amount. The double and triple synergy terms are not simply the differences in rainfall, but rather represent the contributions of the synergistic interactions that occur (Grant and van den Heever, 2014). The focus of this study was on the impacts of combined effects, i.e. the different response of cloud and rain formation. To identify and quantify the involved nonlinearities is out of the scope of the present study and could be pursued in future work. Therefore, we decided not to include an additional section on the factor separation methodology with a figure, but mentioned the main outcome at the end of section 3.1.

*"The spread of the model results and the impact of double and triple synergies was demonstrated so far with precipitation deviations, precipitation intensities, and the normalized ensemble spread. Although the quantitative interpretation of nonlinear interactions is not the main focus of this study, we used the factor separation methodology of Stein and Alpert (1993) in order to understand how aerosols, the shape parameter, and soil moisture may interact synergistically. This methodology is a simple way to show how multiple factors and their nonlinear interactions influence a predicted field and has been applied many times in atmospheric sciences, e.g. for aerosol-cloud-land surface interactions within tropical sea breeze convection (Grant and Heever, 2014) or for the effects of topography, convection, latent, and sensible heat fluxes on Alpine lee cyclogenesis (Alpert, 2011). For the four cases analyzed here, we find that the single impact of changing one parameter has a much weaker response as the double or triple synergies (not shown). Furthermore, all double synergies work to enhance accumulated precipitation, whereas all triple synergistic interactions reduce the precipitation in agreement with findings from Grant and Heever (2014). The triple synergies are greater than the double synergies by a factor of approximately three. Whereas the factor separation for the single impacts is always correlated to the rainfall difference compared to the respective reference run, the double and triple synergy terms rather represent the contributions of the synergistic interactions that occur. We must emphasize that synergy terms may not be meaningful for a field that has a finite range like total precipitation and when individual impacts of one parameter dominate the change of precipitation. Nevertheless, the results from the factor separation and ensemble spread shown before demonstrate the importance of*

*considering synergistic effects for convective-scale predictability. "*

Conclusions:
*"Similarly, the factor separation methodology showed a larger impact of triple synergies to the simulation results compared to double synergies or single impacts which demonstrates the importance of considering synergistic effects for convective-scale predictability."*

As recommended, we changed to title of the manuscript to "Impacts of combined microphysical and land-surface uncertainties on convective clouds and precipitation in different weather regimes".

Finally, the third result section only examines the updraft at 5 km, which is insufficient information to examine/determine convective invigoration. The authors need to consider expanding the updraft analysis to different levels or exclude this section from the paper.
Please see our reply to the second to last comment of the reviewer. We included profiles of updraft velocities for each day where one can see a systematic reduction of updraft velocities across all vertical levels.

The following texts are point-by-point comments.

- Table 1: What is the vertical resolution? Is it varying or constant throughout the column? If varying, what is the vertical resolution near the surface, which could impact the boundary layer processes.
  We use a vertically stretched grid with 14 levels in the lowest 1000 m, which is assumed to be high enough for representing boundary-layer processes. We included this information in Table 1.

- Lines 121–123: Are these CCN concentrations based on the observation? The reasoning for selecting these numbers should be demonstrated clearly with appropriate references.
  There are 4 different concentrations of cloud nuclei available in the Segal-Khain activation. We included another reference to this paper and new reference to observations in central Europe. The text now reads:

  *"Pre-calculated activation ratios stored in look-up tables (Segal and Khain, 2006) are used to compute the activation of CCN from aerosol particles. The condensation nuclei are all assumed to be soluble and follow a bi-model size distribution (Seifert et al., 2012). Using the Segal and Khain (2006) activation, four different values of the number density of CNN ($N_{CCN}$) are available, representing maritime ($N_{CCN} = 100\,cm^{-3}$), intermediate ($N_{CCN} = 500\,cm^{-3}$), continental ($N_{CCN} = 1700\,cm^{-3}$), and continental polluted conditions ($N_{CCN} = 3200\,cm^{-3}$). Typical conditions of central Europe are represented by the continental aerosol assumption (Hande et al., 2016)."*

- Line 156: maritime, intermediate, continental, and polluted are not intuitive and misleading unless these CCN values are based on observations over the maritime and continental part/airmass over the simulation domain. I suggest naming them CCN100, CCN500, CCN1700, and CCN3200, for example. If this is too long, please come up with something more informative. It's tough to remember whether continental is lower or higher than polluted. All the values are different degrees of pollution/aerosol loading.
  We believe that the suggested naming is too long. It might work good in the text, but it is not applicable to use these as labels in our figures. We would therefore like to keep it that way. Moreover, in all figures we have always the same order from maritime (m), to intermediate (i), to contintal (c) and to continental polluted (p). It is our hope that this increasing order is sufficient, same labeling has also been applied in the previous companion study in ACP (Barthlott et al., 2022).

- Line 168: Could you add one or more sentences explaining the definition of convective adjustment time scale?

  We included the formula to compute $\tau$ and also some more text in the manuscript.

- Table 2: The authors need to include the justification for choosing 0.17 and 1.09 as weak strong forcing cases. Unlike weak forcing cases, where the convective adjustment time scales have only 11% difference, the two numbers chosen for the strong instances have more than 146% difference. These could have impacted the dramatic deviation from the reference shown in Figure 5b, first panel. A similar concern goes for Figure 6, rows 3 and 4, where the diurnal behavior looks pretty different between these two forcings compared to similar peaks and shapes shown in rows 1 and 2.

  For the differentiation of strong and weak large-scale forcing, it is only important if the daily mean values of this time scale are below (indicating strong forcing) or above (indicating weak forcing) a threshold of 3 h. Although the strength of large-scale forcing might be stronger on 10 June 2019 than on 17 Aug 2020, the precipitation deviations in Figure 5b are very similar for both strong forcing cases despite their difference in the value of the convective-adjustment time scale. Only for dry soil moisture conditions, the case of 10 June 2019 shows larger negative deviations to the reference run. We believe that this behaviour is more related to the soil moisture-precipitation feedback than to different strengths of the large-scale forcing. Moreover, we did not intend to have similar forcing strengths within each forcing type and find it rather advantageous to have some variability in the forcing strength.

- Line 181: While the agreement between simulated precipitation and radar observations is not shown here, could you elaborate more regarding what radar observations were used and what parameter (e.g., accumulated rainfall, hourly precipitation rate) was chosen to make this comparison?

  We included some more information about the product in the text. It now reads:

  *"We also compared the simulated precipitation to data from the precipitation analysis algorithm RADOLAN (Radar Online Adjustment) which combines weather radar data with hourly surface precipitation measurements of about 1300 automated rain gauges (not shown). For 24-h accumulated precipitation, we find an overall good agreement, even if the precise location of individual convective cells are not always simulated at the right place. However, the model succeeds reasonably well in reproducing the observed cloud and precipitation evolution which implies that these runs serve as a good basis for our combined perturbation experiments."*

- Line 188: Why did you choose "domain-integrated precipitation totals" to represent the precipitation response? The black boxed area in Figure 1 includes land with substantially different orography (the south edge of the domain and the north edge of the domain), water, and coastal regions, which all could show very different precipitation characteristics. Especially under weak synoptic scenarios, the coastal rain process could impact the domain-integrated precipitation.

  We intended to investigate the total precipitation deviation for a representative area in central Europe and chose a box covering most of Germany with only little amounts of sea at the north-western edge. This area comprises flat orography to the north and more complex orography in the center and southern part. A separation of the analysis for different orographic regions was beyond the scope of this study but is ongoing work at our group right now. Moreover, the same evaluation domain was used in Baur et al. (2022) and Barthlott et al. (2022).

- Lines 191–195: Was this soil moisture response linear? Several pieces of literature have shown the nonlinear characteristics of the soil moisture impact on convection. This is also related to the limitation of the current study's design. I understand the computation limitation with many

[Figure]

Figure R.2: Mean precipitation amount as a function of initial volumetric water content (VCW).

simulations considering the non-linearity of soil moisture and other parameters. However, the authors should demonstrate discussion on nonlinear responses found either in this study or in previous studies more clearly in this manuscript. Please check Drager et al. (2022) "A Non-Monotonic Precipitation Response to Changes in Soil Moisture in the Presence of Vegetation" for this comment.

The (non)-linearity of the soil-moisture precipitation feedback was not the main focus of this study and is hard to derive from simulations with three different initial soil moisture fields. However, for each day, 20 realisiations with identical CCN concentration and shape parameter are available with 3 different initial soil moisture fields. We computed the correlation of accumulated mean precipitation to the initial soil moisture and find high mean values for all days:
20160605: r = 0.938; 20180609: r = 0.914; 20190610: r = 0.933; 20200817: r = 0.988

In Fig. R.2, the high correlation is visible for many of the different realizations. We included these sentences in the manuscript:

*"Although this study comprises only three different soil moisture realizations, we tested if there is a linear response of accumulated precipitation to initial soil moisture. Recently, a modeling study by Drager et al. (2022) suggests a new type of rainfall response to soil moisture in which intermediate-moisture soils receive less rainfall than do the driest or wettest soils. This non-monotonic soil moisture–precipitation relationship was found to result from the permanent wilting point's modulation of transpiration of water vapor by plants. Our simulations revealed a monotonic soil moisture–precipitation relationship for all runs under strong synoptic forcing and for 85% of the runs under weak synoptic forcing. Mean correlation coefficients were also high and ranged between 0.914 and 0.988. For more robust results, however, a higher number of soil moisture scenarios as applied in Drager et al. (2022) or Barthlott and Kalthoff (2011) would be necessary."*

- Lines 218–219: These referenced studies all used the COSMO model. And the paper you cited earlier, Marniescu et al. (2021), showed the different convective responses to enhanced aerosol loading resulting from different models. So the authors need to look into other papers that used

the various numerical models and examined aerosol-induced convection changes under different synoptic setups.

The reviewer is right about the fact that two of the three cited papers here are based on results with the COSMO model. We are aware of the fact that ICON is another model and differs from COSMO in many ways, and that different models can produce different responses to enhanced aerosol loading. But COSMO was and ICON still is the operational model used at the German Weather Service. And since both models used the same double-moment scheme and investigated the same geographical domain, we believe that these references can be included here. Furthermore, we only state that the magnitude of the precipitation response is larger for weak forcing than for strong forcing. We believe that mentioning further model intercomparison studies are not necessary for the reader. However, we included a statement indicating that also COSMO results were cited.

*"...which is in agreement with previous findings regarding aerosol–cloud interactions with the COSMO model (Barthlott and Hoose, 2018; Keil et al., 2019) and with ICON (Barthlott et al., 2022). Note that different models may produce different responses to aerosol perturbations, but these studies used the same double-moment scheme for simulating convective episodes over central Europe."*

- Lines 231–234: Is there any way to show this using figures? For example, changes in instability between different simulations? While the explanation authors put here makes sense, it's merely speculation without supporting simulated results. The same goes for Lines 236–244. Where is the supporting evidence for cloud size changes?

  Although the absolute magnitudes of CAPE and autoconversion/accretion can not be compared to each other, we computed their percentage deviations compared to the reference run for this case. We find that they almost balance each other which supports our statement in the text. We included these sentences in the paper:

  *"To support this statement, we calculated the percentage deviations of CAPE and autoconversion/accretion of those model runs to the reference run. We find that the percentage magnitudes are almost identical: CAPE decreases by 11.8 % whereas the warm-rain process increases by 11.5 %. Although these variables cannot be used to quantitatively determine their impact on the total rain amount, it nevertheless supports our hypothesis that the CAPE reduction with drier soils can be compensated by the effects of a strengthened warm-rain process."*

  The part about the cloud sizes was speculative only and is not necessary here as the differing rain amounts can be explained just by the intensity and lifetime of precipitation cells. We removed this part of the text:

  *"The answer to this is twofold: (i) the reduced warm-rain process as a result of a less efficient collision–coalescence process leads to a longer cloud lifetime or ; (ii) stronger rain intensities must compensate the smaller cloud cover. We therefore now analyze the daily cycle of 30 min precipitation rates..."*

- Figure 6: Color shadings, instead of colored lines, are hard for readers to compare responses among different aerosol loadings. Please consider other ways of representing four aerosol loadings for clarity.

  We are aware of the fact that individual model runs cannot be identified here. However, plotting individual lines for each model run would result in 20 different lines which would be even harder to distinguish. We believe that the essential characteristics, namely the higher rain intensities for maritime CCN (blue shadings) and weaker ones for polluted conditions (grey shadings) are still visible.

- Line 256: Since Figure 6 only shows the shape parameter-averaged response, the (WETp0 minus WETp8) is not demonstrated via any figure or table. Could you also consider including different responses as a function of the shape parameter?

  We are sorry about the confusion, but "(WETp0–WETp8)" did not mean a difference, we wanted to list all runs in the polluted wet scenario (shape parameters from 0 to 8). But having read the sentence again, we believe that it does not fit here and removed it:

- Line 284: Please also consider including other references not involving the first author.

  As suggested, we included two additional references.

- Lines 363–365: Please explain why the authors examined this ratio. How does this ratio tell about latent heat release/updrafts? This ratio seems only relevant to the relative dominance of cold rain.

  We agree with the reviewer that the ratio of cold- to warm-rain formation is only revelant for the relative dominance of the cold-rain processes. We therefore moved this part of the analysis to the previous section with the budget calculations of microphysical process rates where it fits much better. New Figure 10 now shows the ratio of cold- to warm-rain formation and new Figure 11 shows only latent heat release and updraft velocities.

- Line 377: the sensitivity to different shape parameters → the sensitivity of w 5km(?) to different shape parameters; since there are sensitivities in the first and second rows as a function of the shape parameter.

  No, this text part is about the sensitivity of latent heat release. We added this information in the respective sentence.

- Lines 383–384: Please include a table or a figure showing the mean updrafts to support these statements. At least, the authors should include the numbers of mean updraft value ranges or distribution.

  This part of the text is still referring to old Figure 9, we included the reference to this figure now in this sentence as well. It now reads:

  To study the impact on the dynamics, we computed spatiotemporal averages of updraft velocities (i.e. only positive values were accounted for) for cloudy grid points defined with a total cloud water content larger than $0.3\,\mathrm{kg\,m^{-2}}$ *(Fig. 11b)*.

- Lines 390–391: Do you mean there is no evidence of convective invigoration (or suppression) throughout all vertical levels? No warm-phase invigoration either? Several recent studies (e.g., Igel and van den Heever, 2021 and references therein) have shown the robustness of the warm-phase invigoration, whereas the cold-phase invigoration is not robust but depends on the environment.

  Our simulations do not show any evidence of convective invigoration in all vertical levels. We computed mean profiles of updraft velocities for cloudy grid points for all members of our ensemble and all analyzed days. The results for a shape parameter of 0 is given in Fig. R.3. We find a robust signal throughout all vertical levels namely a systematic reduction of updraft speeds with increasing CCN concentration. It may be that individual cloud systems show an invigoration, but on average over the evaluation region, we don't find convective invigoration. We added this statement in the text to make clear that all vertical levels show this behaviour and that the level of 5 km is suitable to characterize it.

  *"For this analysis we selected the vertical updrafts at a height of 5 km agl because mean pro-*

[Figure]

Figure R.3: Mean updraft velocities for cloudy grid points having a total cloud water content larger than $0.3\,\mathrm{kg\,m^{-2}}$. Only runs with the default shape parameter of 0 are displayed for better readability.

> *files reveal a systematic response of updraft velocities throughout the entire troposphere and the maximum differences between different aerosol loads occur between 5 and 6 km agl."*

- Line 400: Please check Grant and van den Heever (2014) for how they computed synergistic effects when multiple parameters were perturbed.
  We followed the reviewer's suggestion and used the factor separation methodology as in Grant and van den Heever (2014). Please see our reply to the first comment.

**Additional corrections**

We re-phrased some of the text in order to remove passages which had similarities to previous work. The meaning of the sentences was not changed, therefore the changes are not highlighted in the tracked-changes version.

---

## Author Comment (AC2)

**Responses to the reviewers**

Impacts of combined microphysical and land-surface uncertainties on convective clouds and precipitation

by C. Barthlott, A. Zarboo, T. Matsunobu, and C. Keil                              August 1, 2022
* * *
We thank both reviewers for reading the manuscript and providing detailed comments. We have carefully considered all comments and changed the manuscript accordingly. Please find below our responses in blue.

**Reviewer 2**

This study investigated the model uncertainties associated with three factors: soil moisture, CCN concentration, and the shape factor of cloud drop size distribution. Quite a few similar studies have been conducted recently, but none applied such a three- parameter combination. The results showed significant spreads caused by the two microphysical parameters, and the soil moisture factor also enhances the spread. But the significance of these factors compared to many others in the model is unclear. The manuscript can be enriched if the suggestions in the major comments below can be considered.

Note: I have posted a preliminary version of the review, and sorry for double-posting some of the comments.

**Major comments**

Introduction and methodology:

1. As the authors stated, model uncertainties exist in many physical schemes and dynamics, including initial/boundary conditions (IC/BCs). There can be numerous combinations of such uncertainty sources. Can the authors explain why it is essential to consider the combination of soil moisture and cloud microphysics compared to other possible combinations?

   Soil moisture influences the initiation and development of convective systems and many papers in recent years documented the complex land-surface precipitation relationship, e.g. the review article of Liu et al. (2022). Although the influence of soil moisture on convective precipitation follows some definite rules, the relationship is complex and the direction of the soil moisture influence on convective systems and precipitation can vary. As soil moisture controls the partitioning of the available energy at the ground into sensible and latent heat, the structure of the planetary boundary-layer is significanty affected. This is not only important for convection initiation, but also for already existing convective systems which depend upon the existence of CAPE. In the operational data assimilation system of the German Weather Service (DWD), soil moisture is also one of the disturbed parameters. On the other hand, microphysical uncertainties are not yet included operationally, but are shown to have a non-negligible impact in this work and also in a prior study (Barthlott et al., ACP, 2022). We do not state that the combination of soil moisture and microphysical uncertainties are essential to consider or superior to other ways of generating an ensemble, but we believe that the spread of the results of our method looks promising and that the comparison with other sources of uncertainty should be done. First efforts in this direction have been performed by Matsunobu et al. (WCD, 2022). As soil moisture and microphysical uncertainties both influence cloud development at different stages and their individual impact has been demonstrated in many recent papers (Schneider et al. 2019; Keil et al. 2019), we believe that our method of combined uncertainties is well suited to be compared to other sources of uncertainty.

We added this statement in the introduction:

*"We choose these uncertainties because (i) their individual impact was documented in many recent studies and (ii) all have an impact on the life cycle of convection at different stages from its initiation to the decay."*

2. Similarly, there are many uncertainties in cloud microphysical parameterizations. How are these factors considered in this study? Can the authors justify why they focused only on uncertainties in NCN and CDSD parameters? Also, the authors mentioned many uncertainties related to aerosol-cloud interactions (lines 45-76). Can these uncertainties be represented by perturbing the NCN?

We agree with the reviewer that there are many uncertainties in cloud microphysical parameterizations. However, it was not the goal of this study to investigate many different microphysical uncertainties as our concept of combined uncertainties based on soil moisture, CCN concentration, and shape parameter already yields to an ensemble size of 60. Moreover, as the analysis of microphysical process rates shows, many different processes are affected directly and indirectly. We therefore believe that our uncertainties affect many microphysical pathways although only warm-rain processes are influenced directly. Including more microphysical uncertainties is out of the scope of the present study. We included a reference to Wellmann et al. (2020) who investigated more microphysical uncertainties, but only for idealized simulations. We added this sentence at the end of section 2.1:

*"The reference run would therefore be labeled as run REFc0. Including more microphysical uncertainties (e.g. ice nucleating particle concentration, hydrometeor sedimentation, or ice multiplication) as in idealized simulations by Wellmann et al. (2020) could be considered in the future, but were not performed at the moment due to the high number of possible combinations."*

3. Line 122-123: There is a difference between NCN and NCCN (CN stands for condensation nuclei and CCN for cloud condensation nuclei). For polluted continental conditions, the value of 3200 cm-3 seems to be too low for NCN (should be tens of thousands or more) but fine for NCCN. The values used for other conditions should also be justified or, at least, provide a reference.

Thanks for pointing that out, we corrected the text, it now reads:

*"Pre-calculated activation ratios stored in look-up tables (Segal and Khain, 2006) are used to compute the activation of CCN from aerosol particles. The condensation nuclei are all assumed to be soluble and follow a bi-model size distribution (Seifert et al., 2012). Using the Segal and Khain (2006) activation, four different values of the number density of CNN ($N_{CCN}$) are available, representing maritime ($N_{CCN} = 100\,cm^{-3}$), intermediate ($N_{CCN} = 500\,cm^{-3}$), continental ($N_{CCN} = 1700\,cm^{-3}$), and continental polluted conditions ($N_{CCN} = 3200\,cm^{-3}$). Typical conditions of central Europe are represented by the continental aerosol assumption (Hande et al., 2016)."*

4. The shape parameter $\nu$ is also important for other hydrometeors. In fact, the variation in $\nu$ may be even more prominent for precipitation particles according to some triple-moment schemes. What is the reason for perturbing only $\nu$ of cloud drops?

As already mentioned above, not all microphysical uncertainties can be assessed in our study. One reason for taking the shape parameter of the CDSD was the fact that it is not well constrained by observations and many different values exist in the literature. The results of our study also shows the indirect effect on rain and ice formation via the large differences in microphysical process rates. Also the already quite high number of possible perturbations when using the three uncertainties analyzed here inhibits us from considering even more uncertainties at the moment.

Results:

1. The model "spread" is one of the key foci of this study. Yet, the discussion on the normalized standard deviation (indicating the spread) is too brief and does not provide much scientific insight.
   Please see our reply to the next point.

2. I would like to see a more quantitative comparison of spreads from the three sensitivity factors (i.e., soil moisture, CCN, and shape factor. This allows the reader to judge which factors are more important for the consideration of ensemble members.
   As suggested by the reviewer we performed such a separate analysis of the three impact factors and included the temporal evolution for all days in new Figure 7. The time series of the total ensemble spread is based on 60 members whereas for the individual contributions, the ensemble size is smaller. E.g. for the soil moisture sensitivity, there are only 3 members with identical CCN and shape parameters. The ensemble spread was then averaged over all 20 3-member ensembles. Thus we applied bootstrapping to randomly pick 3 suitable combinations and repeated that procedure 100 times. Due to the smaller size of the sub-ensembles, the individual contribution from CCN, shape parameter, and soil moisture may not be fully reliable because the normalised spread is actually bounded by the square root of ensemble size and all spreads become very similar when ensemble size is very small. The important point is the total spread of the 60-member ensemble (black lines in new Fig. 7) which lies in a similar range as an operational ensemble for a high-impact weather period in 2016. Furthermore, we believe that the model spread is also visible in the deviations of accumulated precipitation (Fig. 5) and the precipitation rates (Fig. 6).

   We included the new Figure 7 together with this text:

   *" Beside an ensemble spread based on all 60 members, we also computed the spread induced by soil moisture, CCN, and the shape parameter individually. As only three soil moisture regimes are available for each identical CCN concentration and shape parameter, we used the bootstrapping method to randomly pick between different suitable combinations to calculate their normalised spread. This procedure was repeated 100 times. The results show that the area-averaged local precipitation variability introduced by varied CCN concentrations and shape parameters is rather similar (Fig. 7). This finding holds true for all days irrespective of the synoptic-scale forcing. For both uncertainties, the variability increases rapidly already in the first hours of the forecast, followed by a rather constant plateau until a further increase occurs in the afternoon at the peak of convective activity (see rain rates in Fig. 6). In contrast to that, the variability due to soil moisture reveals a weaker increase early in the simulation and reaches similar high values (or even higher ones on 9 June 2018) as CCN and shape parameter variability only around noon. In the afternoon, a similar weak increase is simulated as in the other types of uncertainty. Later, soil moisture variability remains slightly below the ones from CCN and shape parameter. For a high-impact weather period of 2016, Keil et al. (2019) found that the spread induced by soil moisture was slightly larger than the one induced from different CCN concentrations in the afternoon. However, in their study soil moisture was perturbed by applying high-, low- and bandpass filters to introduce surface perturbations which is different from our approach of using a soil moisture bias. Figure 7 further reveals that..."*

3. Line 249-250: "The higher the CCN concentration, the lower are the rain intensities." This seems to be a warm-rain characteristic. But, apparently, the studied systems are mostly cold-rain dominant (lines 443-444). For mixed-phase convective systems, higher aerosol concentrations often lead to stronger rain intensity (cf. Tao et al. 2012, etc.). It will be nice to compare the results

here with other relevant studies.

The results of our study are based on mixed-phase convective systems, but the cold-rain contribution is always dominant, at least when integrated over the entire day. The previous work of Barthlott et al. (ACP, 2022) documented dominant warm-rain contributions only for very weak rain intensities probably at very early stages of the precipitation formation in clouds. There are a number of studies who also find a precipitation reduction with increasing CCN concentrations and we mentioned at least some of these in the manuscript in the introduction:

*"However, the impact of aerosols on convective precipitation has been shown to differ between cloud types, the aerosol regime, and environmental conditions (e. g. Seifert and Beheng, 2006b; Khain et al., 2008; van den Heever et al., 2011; Tao et al., 2012; Barthlott et al., 2017)."*

We added this text in section 3.1 discussing the decreasing precipitation totals with increasing CCN:

*"The validity of the convection invigoration mechanism proposed in Rosenfeld et al. (2008) is still open and many studies documented a decrease of total precipitation with increasing aerosol concentrations (e.g. Tao et al., 2012; Storer and van den Heever, 2013). Using idealized simulations, Grant and van den Heever (2015) showed that the influence of aerosols varies inversely with storm organization and Fan et al. (2009) found that vertical wind shear qualitatively determines whether aerosols suppress or enhance convective strength. "*

4. Figure 7. The tendency of TQG change with $\nu$ is different for maritime CN compared to other CN types for cases 2018 and 2020. Some inconsistencies also exist in the 2016 case. Is there any explanation?

   We mentioned in the manuscript that the case with systematic graupel reduction with increasing CCN or shape parameter is the one with the highest integrated graupel content (10 June 2019), the remaining cases had much lower graupel contents. A closer look at the microphysical process rates reveals that the response of graupel follows mostly the one from graupel/hail riming with cloud droplets (Fig. R.1). We also see that riming with cloud droplets is dominating the riming with rain droplets. For maritime CCN conditions, there is no large sensitivity to the shape parameter, but for polluted conditions and already narrow CDSD, the impact is higher for the three cases with lower overall graupel contents .

   We added this remark in the manuscript:

   *"Some of the cases show decrease in graupel mass for maritime CCN conditions and an increase for higher CCN concentrations. This can be attributed to graupel/hail riming with cloud droplets which increases with larger shape parameters for already more narrow size distributions (not shown)."*

Conclusion:

1. It is dangerous to make a conclusion based on only four cases. Large differences can be observed between the two weak cases or between the strong cases, which may suggest that other cases may behave distinctively differently and even produce results that disagree with the conclusions stated here. Furthermore, the uncertainties in the studied parameters may vary if you choose different initial/boundary conditions, physics schemes, or grid resolutions. The authors should at least try to tone down a bit on the certainty of their findings.

   We are aware of the fact that general conclusions cannot be based on 4 cases only. Altogether, we constructed our 60-member ensemble for 8 cases but only mentioned 4 of them in the text. We made this selection because the sensitivity was mostly similar among the cases and the detailed analyses of the microphysical processes for every day would only lengthen the manuscript

[Figure]

Figure R.1: Spatiotemporal averages of graupel/hail riming with cloud droplets (left) and with rain droplets (right) from the respective reference run in kg/(kg 24 h).

unnecessarily.

2. Perhaps the authors can make a quantitative comparison of the spread caused by each factor by preparing a table summarizing the relative spreads (standard deviation).
We now included such a quantitative comparison, please see our reply to comment Results no. 2.

3. There are quite a few similar studies with multiple-factors analyses. Because of the numerous possible combinations of uncertainty factors, it will be nice to see some comparisons on the spread/uncertainty with previous studies.
We included a couple of new citations in the introduction:

*"Other studies with multiple-factor analyses exist mostly for idealized setups, e.g. for investigating the impact of environmental conditions and microphysics on the forecast uncertainty of deep convective clouds and hail using an emulator approach by Wellmann et al. (2020), for investigating aerosol–cloud–land surface interaction within tropical sea breeze convection (Grant and Heever, 2014) or investigating the relative sensitivity of a tropical deep convective storm to changes in environmental and cloud microphysical parameters (Posselt et al., 2019). Using the Morris one-at-a-time (MOAT) method for simultaneous perturbations of numerous parameters, Morales et al. (2019) explored the sensitivity of orographic precipitation within an environment of an atmospheric river. "*

and these sentences in the Conclusions:

*"To our knowledge, only Grant and Heever (2014) and Baur et al. (2022) studied synergistic effects of aerosols and soil moisture so far. Grant and Heever (2014) conducted idealized cloud-resolving simulations of tropical sea breeze convection and found precipitation reductions by over 40% and 50% for the most extreme perturbations. The realistic convection-resolving simulations of Baur et al. (2022) were conducted for a single case study only, but they found a similar sensitivity of precipitation deviations as in this study (-23% and +10%)."*

**Minor comments**

1. Line 5: 60 member ensemble → 60-member ensemble (same in other places of the text)
done

2. Line 12-13: rain water → rainwater
We would like to keep the spelling as it is, since it was written the same way in a previous companion study in ACP:

*Importance of aerosols and shape of the cloud droplet size distribution for convective clouds and precipitation* by Christian Barthlott, Amirmahdi Zarboo, Takumi Matsunobu, and Christian Keil, Atmos. Chem. Phys., 22, 2153–2172, https://doi.org/10.5194/acp-22-2153-2022, 2022

3. Line 14: strong, but → strong but
done

4. Line 14: non systematic → non-systematic
done

5. Line 15: which → , which
done

6. Equation (1): Since the microphysics scheme used is double moments with $A$ and $\lambda$ as varying coefficients, $\nu$ and $\mu$ must be specified. The value for $\mu$ was never mentioned. If $\mu$ was set to 1,

then just omit it in the equation.
We added this sentence in the manuscript:

*"The dispersion parameter is kept constant in all simulations ($\mu = 1/3$)."*

7. Figures 5, 7-9: These figures are quite complicated. More details (e.g., what is NU) are needed in the caption to assist the readers in understanding the arrangements.
We are aware of the fact that these figures are complicated and already tried to give as much information in the caption as possible. Although the shape parameter is mentioned there, NU was not. We now included that and hope that the reader has now all necessary information. The caption of new Figs. 8–11 now contain information that the arrangement of the data points is exactly as in Figure 5.

8. Line 215: applies for → applies to
done

**Preliminary version of review:**

As the authors stated, model uncertainties exist in many physical schemes and dynamics, including initial/boundary conditions (IC/BCs). There can be numerous combinations of such uncertainty sources. Can the authors explain why it is essential to consider the combination of soil moisture and cloud microphysics compared to other possible combinations?
Please see our reply to major comment 1 above.

A similar question: there are many uncertainties in cloud microphysical parameterizations. How are these factors considered in this study? Can the authors justify why they focused only on uncertainties in N_CCN and CDSD parameters? Also, the authors mentioned many uncertainties related to aerosol-cloud interactions (lines 45-76). Can these uncertainties be represented by perturbing the N_CCN?
Please see our reply to major comment 2 above.

The discussion on the normalized standard deviation is too brief and does not provide much scientific insight.
Please see our reply to Results comment 1 and 2 above.

I would like to see a more quantitative comparison of spreads from the three sensitivity factors (i.e., soil moisture, CCN, and shape factor.
Please see our reply to Results comment 1 and 2 above.

**Additional corrections**

We re-phrased some of the text in order to remove passages which had similarities to previous work. The meaning of the sentences was not changed, therefore the changes are not highlighted in the tracked-changes version.

---

## Author Response (AR2)

**Responses to the reviewers**

Impacts of combined microphysical and land-surface uncertainties on convective clouds and precipitation in different weather regimes

by C. Barthlott, A. Zarboo, T. Matsunobu, and C. Keil                    August 17, 2022
* * *
We thank both reviewers for reading the revised manuscript again. We have carefully considered all remaining comments and changed the manuscript accordingly. Please find below our responses in blue.

**Reviewer 1**

The authors have responded satisfactorily to most of my previous comments, except for major comments #2 and 4 (i.e., Why choose the shape parameter to represent microphysical uncertainties?). The way it was written in the text gives the reader the impression that the selection of microphysical parameters is arbitrary. My main point is whether the uncertainty associated with the shape factor is substantially significant compared to other microphysical uncertainties (in producing spreads). I suggest the authors at least provide a reference indicating that the uncertainty associated with the shape parameter for cloud drop size distribution is sufficiently representative.

We tried to justify the choice of the shape parameter in the introduction as the width of the CDSD is not well constrained by measurements and a wide range of values (between 0–14) based on cloud type and environmental conditions were reported. We also mentioned variables that are directly and indirectly affected by the shape parameter together with references to idealized (Igel and van den Heever, 2017) and realistic simulations (Barthlott et al., 2022). In these papers, the potentially large impact of the shape parameter on the simulation results is documented. E.g. Barthlott et al. (2022) found that the increase in the shape parameter can produce almost as large a variation in precipitation as a CCN increase from maritime to polluted conditions. Furthermore, in the widely used Thompson-Eidhammer cloud microphyics scheme, the shape parameter is one of the stochastically perturbed parameters. We included references to two recent papers and hope that our choice to simultaneously perturb the shape parameter together with CCN concentrations and soil moisture is sufficiently motivated now.

*"Furthermore, the shape parameter is one of the stochastically perturbed parameters in the widely used Thompson-Eidhammer cloud microphyics scheme and recent model results indicate a suitability of this parameter for generating ensembles at the convective scale (Griffin et al., 2020; Thompson et al., 2021)."*

An additional comment: The precipitation response to soil moisture may depend on whether the cloud formation location is energy-limited or moisture-limited for the particular cases. I suggest describing where the studied conditions (including the perturbed soil moistures) are placed on the Budyko-curve plot (i.e., do they stay mostly in one of the regimes or spread widely across them?). This will give the readers an indication of the applicability of the results for different regions. Also, it might help explain the nonlinearity exhibited in the precipitation response to soil moisture shown in various figures.

We agree with the reviewer that such an analyses might be helpful, but unfortunately the evapotranspiration was not part of the model output as we focussed more on microphysical process rates. Thus, we cannot generate a Budyko-curve plot for our cases. However, we already mentioned the mean relative water content at initial time for the runs with reference CCN concentration and shape parameter in section 3.1. As suggested by the reviewer, we now added the values of the runs with perturbed soil moisture as well. Instead of mentioning them in the text, we now added these values

to Tab. 2 with the overview of the cases. We believe that the inclusion of the RWC to that table will help to characterize our cases with respect to soil moisture initialization. The table now reads:

Table 1: List of cases with convective adjustment time scale $\tau$ *and mean initial relative water content RWC for the three soil moisture scenarios*.

| Synoptic-scale forcing | Date | $\tau$ (h) | *RWC (DRY/REF/WET) (%)* |
|---|---|---|---|
| weak | 5 June 2016 | 5.22 | *55/73/86* |
| weak | 9 June 2018 | 4.65 | *28/37/46* |
| strong | 10 June 2019 | 0.17 | *24/33/41* |
| strong | 17 August 2020 | 1.09 | *25/37/42* |

**Reviewer 2**

I appreciate the authors' efforts to address reviewers' comments and questions. One minor thing to note is Author's response on pages 12–13, item 3. There is this recent paper besides Grant and van den Heever (2014) and Baur et al. (2022). Please find this: https://acp.copernicus.org/preprints/acp-2021-693/

Thank you for pointing that out, we added this reference to our manuscript. It now reads:

To our knowledge, only Grant and Heever (2014), *Park and van den Heever (2021)* and Baur et al. (2022) studied synergistic effects of aerosols and soil moisture so far. Grant and Heever (2014) conducted idealized cloud-resolving simulations of tropical sea breeze convection and found precipitation reductions by over 40% and 50% for the most extreme perturbations. *Park and van den Heever (2021) have performed two large idealized 130-member ensembles that represent different initial conditions typical of tropical sea breeze environments in which they simultaneously perturbed six atmospheric and four surface parameters. Comparisons of the clean and polluted ensembles demonstrated that aerosol direct effects reduce the incoming shortwave radiation, as well as the outgoing longwave radiation, within the polluted ensemble and that enhanced aerosol loading results in a weakening of the convection initiated along the sea breeze front.* The realistic convection-resolving simulations of Baur et al. (2022) ...